# A general interfacial-energetics-tuning strategy for enhanced artificial photosynthesis

Tian Liu[1,2], Zhenhua Pan [3] ✉, Kosaku Kato[4], Junie Jhon M. Vequizo [5], Rito Yanagi[6,7], Xiaoshan Zheng[1], Weilai Yu [8], Akira Yamakata[4], Baoliang Chen [1], Shu Hu [6,7], Kenji Katayama[3] & Chiheng Chu [1] ✉

The demands for cost-effective solar fuels have triggered extensive research in artificial photosynthesis, yet the efforts in designing high-performance particulate photocatalysts are largely impeded by inefficient charge separation. Because charge separation in a particulate photocatalyst is driven by asymmetric interfacial energetics between its reduction and oxidation sites, enhancing this process demands nanoscale tuning of interfacial energetics on the prerequisite of not impairing the kinetics and selectivity for surface reactions. In this study, we realize this target with a general strategy involving the application of a core/shell type cocatalyst that is demonstrated on various photocatalytic systems. The promising $H_2O_2$ generation efficiency validate our perspective on tuning interfacial energetics for enhanced charge separation and photosynthesis performance. Particularly, this strategy is highlighted on a $BiVO_4$ system for overall $H_2O_2$ photosynthesis with a solar-to-$H_2O_2$ conversion of 0.73%.

Solar energy is deemed as a key solution to the increasing global demands on clean energy and related climate issues[1–3]. To tackle the spatiotemporal fluctuations of solar radiation, a promising strategy is to capture solar energy in storable and transportable solar fuels (e.g., $H_2$ and $H_2O_2$) by artificial photosynthesis[4,5]. Among primary photosynthetic systems, the particulate photocatalyst is considered as the most cost-effective one because of its high simplicity and scalability, yet its energy conversion efficiency still needs improving for viable applications[6,7].

In pursuit of high-performance particulate photocatalysts, a key challenge is to improve charge separation[8]. Enhancing charge separation demands exquisitely tuning interfacial energetics between reduction and oxidation sites, and in the meantime not impairing the kinetics and selectivity for surface reactions. For instance, most particulate photocatalysts are n-type semiconductors with a Fermi level close to their conduction band, and the photocatalyst/cocatalyst interfaces typically form Schottky junctions, with upward band bending on both the reduction and oxidation sites (Fig. S1a)[9]. The Schottky junction at the oxidation sites facilitates hole migration to the surface reaction sites, favoring charge separation. In contrast, the Schottky junction at the reduction sites impedes electron migration and traps holes, undermining charge separation. Further, it reduces the band offset potential between the reduction and oxidation sites ($\Delta V$), the fundamental driving force for charge separation (Fig. S1a)[10,11]. Such detrimental effects of a high Schottky barrier at the reduction sites have been verified in the previous studies (e.g., Pd-loaded $BiVO_4$ for $H_2O_2$ generation[12], and Pt-loaded GaN or $LaTiO_2N$ for $H_2$ evolution[13,14]). Therefore, the Schottky barrier at reduction sites is a major hurdle limiting charge separation and further artificial photosynthesis performance. Yet, how to lower the Schottky barrier at reduction sites remains a critical challenge for particulate photocatalysts (Fig. S1b).

[1]Faculty of Agriculture, Life, and Environmental Sciences, Zhejiang University, 310058 Hangzhou, China. [2]Suzhou Institute for Advanced Research, University of Science and Technology of China, 215000 Suzhou, China. [3]Department of Applied Chemistry, Faculty of Science and Technology, Chuo University, Bunkyo, Tokyo 112-8551, Japan. [4]Faculty of Natural Science and Technology, Okayama University, Kita-ku, Okayama, Japan. [5]Research Initiative for Supra-Materials, Shinshu University, Nagano-shi, Nagano 380-8553, Japan. [6]Department of Chemical and Environmental Engineering, Yale University, New Haven, CT 06511, USA. [7]Energy Sciences Institute, Yale University, West Haven, CT 06516, USA. [8]Department of Chemical Engineering, Stanford University, Stanford, CA 94305, USA. ✉e-mail: zhenhua.20y@g.chuo-u.ac.jp; chuchiheng@zju.edu.cn

To tune interfacial energetics while preserving the kinetics and selectivity of surface reactions, a possible strategy is to apply a binary cocatalyst composed of two parts for the respective function. Such a strategy has been applied in the photoelectrochemical water splitting systems[15,16]. For example, Esposito et al. applied a Ti/Pt double-layer cocatalyst on a p-Si photocathode for enhancing $H_2$ evolution. In this system, the Ti layer formed a desirable junction with p-type Si for charge separation, while the Pt layer on the Ti layer efficiently catalyzed $H_2$ evolution[16]. However, applying this strategy on a particulate photocatalyst is challenged by its complicated configuration where reduction and oxidation sites are assembled on the same surface with a nanoscale distance. Under such an intricate condition, the energetics at reduction sites need to be tuned selectively and locally without affecting those at the oxidation sites, otherwise the overall charge separation will be affected. These challenges have impeded the practice of interfacial energetics tuning for enhancing charge separation in a particulate photocatalyst.

Here, we report the interfacial energetics tuning in a particulate photocatalyst for photosynthesis of $H_2O_2$, an emerging liquid solar fuel substitute for gaseous $H_2$[17]. Organic semiconductors have been developed for photocatalytic $H_2O_2$ generation with reported solar-to-$H_2O_2$ conversion (STH) up to 1.5%[18–21]. Yet, organic materials have a potential stability concern because photocatalytic $H_2O_2$ generation is inevitably accompanied by hydroxyl radical (•OH) generation ($H_2O_2 + h\nu \rightarrow 2•OH$ or $H_2O_2 + e^- + H^+ \rightarrow •OH + H_2O$), which can damage organic structures[22,23]. Our group recently developed an inorganic $BiVO_4$ photosynthesis system with a STH of 0.29%. The apparent quantum yield (AQY) of the system is 5.8% at 420 nm, indicating that majority of the charge carriers recombined during reaction and charge separation needs further tuned for higher photocatalytic performance[12]. In this study, we constructed a Ag/Pd binary cocatalyst with core/shell structure on the {010} reduction facets of $BiVO_4$. $BiVO_4$ is a visible-light-responsive photocatalyst promising for long-term $H_2O_2$ generation because it is resistive to •OH (a potent oxidant inevitably formed during $H_2O_2$ generation) compared to the organic counterparts[24–26]. The Pd shell served to steer the oxygen-reduction pathway to two-electron process for $H_2O_2$ synthesis[27–29], while the Ag core served to lower the Schottky barrier between $BiVO_4$ and Pd. In-depth time-resolved spectroscopic investigations and electrical numerical simulations on photogenerated charge carriers demonstrate that Ag core significantly lowered the Schottky barrier at the {110} reduction facets and enhanced charge separation in $BiVO_4$. Without using any sacrificial agent, the rationally designed $BiVO_4$ photocatalyst achieved a new record among inorganic photocatalysts (Table S1). Even comparing with the sacrificial system, $CoO_x/BiVO_4/$(Ag/Pd) has a comparable $H_2O_2$ production yield (Table S1). The feasibility of core/shell cocatalyst construction for interfacial-energetics-tuning was further demonstrated on $C_3N_4$ and $TiO_2$, demonstrating its high generality for enhancing charge separation and performance in photosynthesis systems.

## Results and discussion

### Interfacial energetics tuning through applying core/shell type cocatalyst

We first synthesized faceted $BiVO_4$ particles using a solid-liquid-reaction method. The X-ray diffraction (XRD) pattern of $BiVO_4$ particles matched well with that of monoclinic $BiVO_4$, with {010} and {110} facet peaks located at 30.6° and 18.7°, respectively (Fig. S2). Scanning electron microscope (SEM) images (Fig. S3) show that the $BiVO_4$ particles exhibit a decahedron structure with clear facets. Following previous studies[12,30], the top/bottom and side facets are assigned to {010} reduction facets and {110} oxidation facets, respectively.

We then loaded reductive and oxidative cocatalysts onto the corresponding facets of $BiVO_4$ via stepwise photodeposition. $CoO_x$ served to enhance water oxidation and was deposited onto {110} facets via photooxidation of $Co^{2+}$ ions ($CoO_x/BiVO_4$, Fig. 1a). SEM image shows that $CoO_x$ particles are loaded on the surface of $BiVO_4$ (Fig. S4a). Energy-dispersive X-ray spectroscopy (EDS) elemental mapping and line profile indicate that Co signal on {110} facets is much higher than that on {110} facets, confirming the selective loading of $CoO_x$ on the {110} facets (Fig. 1b). The particles exhibit prominent Co 2p X-ray photoelectron spectroscopy (XPS) peaks (Fig. S5). The Co $2p_{3/2}$ peak can be deconvoluted to a $Co^{2+}$ peak at 781.6 eV and a $Co^{3+}$ peak at 780.6 eV, suggesting that the valence of Co was in-between +2 and +3 and therefore it can be denoted as $CoO_x$.

Selective deposition of Ag and Pd on the {010} facets of $BiVO_4$ was performed via stepwise photoreduction of $Ag^+$ and $PdCl_4^{2-}$ to form a core/shell structure ($CoO_x/BiVO_4/$(Ag/Pd), Fig. 1a). SEM images clearly show Ag (Fig. S4b) and Ag/Pd nanoparticles on the {010} facets (Fig. S4c, d). Prominent Ag 3d and Pd 3d XPS peaks demonstrate the successful loading of metallic Ag/Pd cocatalysts, with the Ag 3d peaks at 368.1 eV and 374.2 eV assigned to the Ag $3d_{5/2}$ and Ag $3d_{3/2}$, respectively (Fig. S6), and the Pd 3d peaks at 335.1 eV and 340.6 eV assigned to Pd $3d_{5/2}$ and Pd $3d_{3/2}$, respectively (Fig. S7). The minor $Pd^{2+}$ peak at 337.0 eV is attributed to PdO due to the partial oxidation of Pd (Fig. S7).

Consistent with elemental mapping observations, line profile confirms the facet-selective loading of cocatalysts, with 3.0-fold stronger Co signals on the {110} facets compared to those on the {010} facets, as well as 6.1- and 3.3-fold stronger Ag and Pd signals on the {010} facets compared to those on the {110} facets (Fig. 1b). Notably, the Ag and Pd signals in line profile overlap in distance (Fig. 1b), indicating the co-loading of Ag and Pd nanoparticles on the same sites. The facet-selective loading of Co, Ag and Pd was further confirmed by STEM-EDS mapping from top view and side view of $CoO_x/BiVO_4/Ag$ (Fig. S8) and $CoO_x/BiVO_4/Pd$ (Fig. S9).

We further analyzed the structure of the Ag/Pd nanoparticles by scanning transmission electron microscopy-energy dispersive spectrometer (STEM-EDS). Top view element mapping images show a great overlap between Ag and Pd, with the distribution area of Pd slightly larger than Ag, indicating the formation of core (Ag)/shell (Pd) structure (Fig. 1c). Such a structure is further displayed by a distinct Pd shell slightly larger than Ag core as shown in the side view images (Fig. 1d). These results demonstrate the core/shell structure of the Ag/Pd cocatalyst, formed via selective deposition of Pd on Ag nanoparticles where photoexcited electrons accumulated (Fig. 1a).

The effectiveness of loading Ag on tuning interfacial energetics was investigated by comparing the overall Schottky barrier heights of $BiVO_4$, $BiVO_4/Pd$, $BiVO_4/Ag$, and $BiVO_4/$(Ag/Pd) using ultraviolet photoelectron spectroscopy (UPS). Here, the overall Schottky barrier heights are quantified by the distance between valence band and the Fermi level position, ($E_v - E_f$)[31]. The ($E_v - E_f$) of $BiVO_4$, $BiVO_4/Ag$, $BiVO_4/$Pd, and $BiVO_4/$(Ag/Pd) were determined to be 2.22, 2.35, 2.25 and 2.36 eV, respectively (Fig. 1e). Since Pd has a high work function (5.6 eV), $BiVO_4/Pd$ built up a high Schottky barrier. In comparison, $BiVO_4/Ag$ exhibits a lower Schottky barrier attributed to the low work function of Ag (4.3 eV). Such a low Schottky barrier was maintained on $BiVO_4/$(Ag/Pd) with a direct contact between Ag and $BiVO_4$ (Fig. 1f). Consistent with UPS results, the binding energies of Bi and V in XPS spectra follow the order of $BiVO_4/Ag < BiVO_4/Pd < BiVO_4$ (Fig. S10), indicating that Ag on {010} facets reduced excitation energy of $BiVO_4$ valence electrons by lowering the Schottky barrier. Notably, both the UPS and XPS analyses are based on overall energetics of $BiVO_4$ rather than the local one at the {010} facets. Therefore, the impact of Ag on the Schottky barrier at {010} facets should be even higher than UPS observations (confirmed by electrical simulation results, see below). While it is desirable to directly compare the local Schottky barriers at {010} facets with recently developed potential-sensing electrochemical atomic force microscopy in Boettcher group, its application on a particulate photocatalyst is still under progress[32,33].

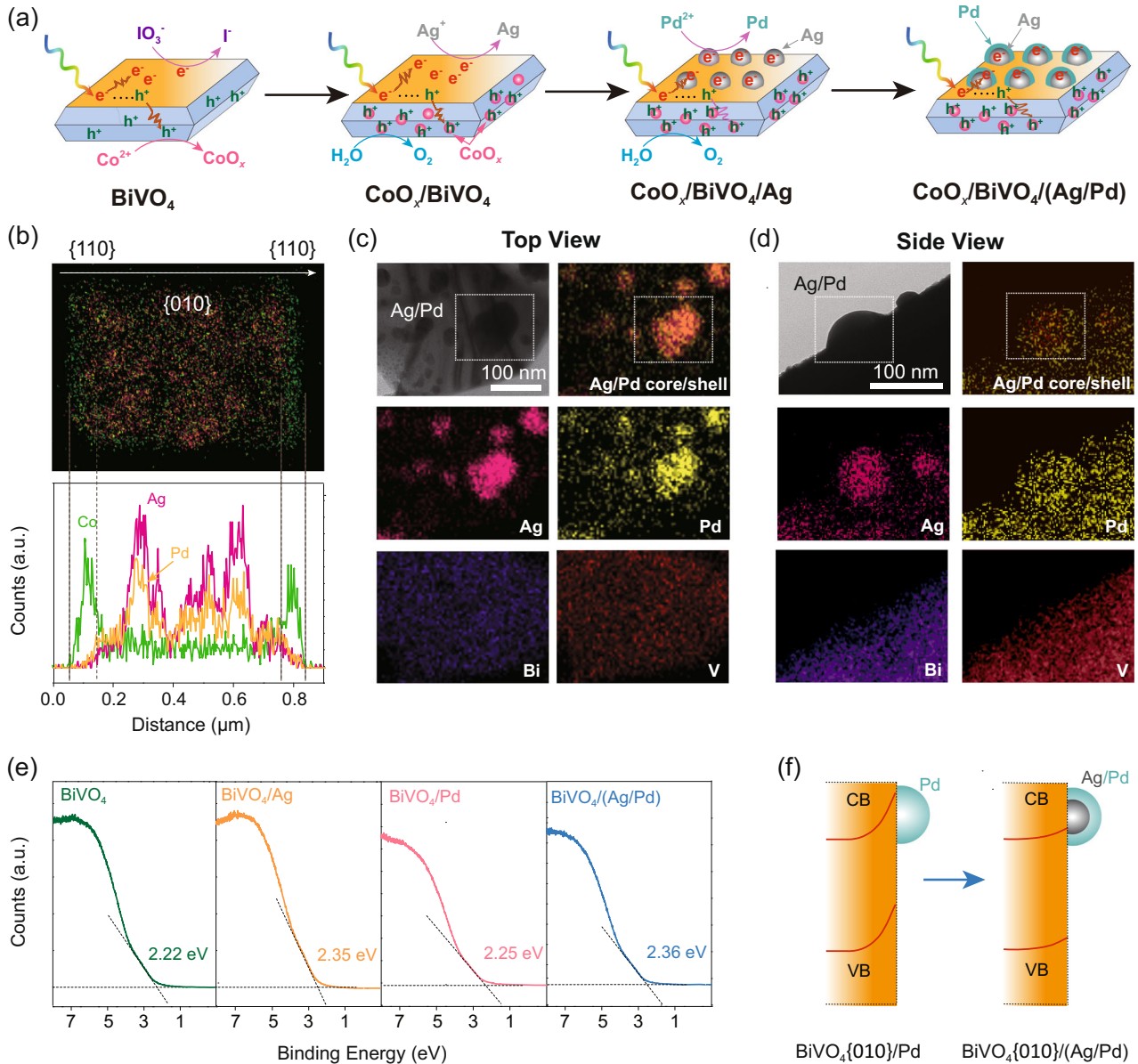

**Fig. 1 | Facet-selective loading of cocatalysts on BiVO₄ and interfacial energetics tuning with Ag/Pd core/shell cocatalyst. a** Stepwise and facet-selective photodeposition of Co, Ag, and Pd on BiVO₄. **b** Energy-dispersive X-ray spectroscopy (EDS) elemental mapping and line profile along with the white arrow of CoOₓ/BiVO₄/(Ag/Pd). We increased Co (2 wt%), Ag (1 wt%), and Pd (1 wt%) loadings for more clear observation. **c, d** Scanning transmission electron microscopy (STEM)-EDS elemental mapping of Ag/Pd particles loaded on BiVO₄. We increased Ag (1 wt%) and Pd (1 wt%) loadings for more clear observation. **e** Ultraviolet photoelectron spectroscopy (UPS) spectra of BiVO₄, BiVO₄/Ag, BiVO₄/Pd, and BiVO₄/(Ag/Pd). **f** Schematic illustration of {010} reduction facet interfacial energetics tuning through Ag/Pd core/shell cocatalyst construction on BiVO₄.

## Enhanced photocatalytic H₂O₂ generation with interfacial energetics tuning

We first compare the photocatalytic H₂O₂ generation performance of different particulate photocatalysts under visible light irradiation (λ > 400 nm) in pure water. The performance of CoOₓ/BiVO₄/(Ag/Pd) was optimized by adjusting the loading amount of Co, Ag, and Pd to 0.3 wt%, 0.08 wt%, 0.4 wt%, respectively (Fig. S11–13). Pd loading (i.e., BiVO₄/Pd) enhanced H₂O₂ generation by 7.9-fold compared to bare BiVO₄ (Fig. 2a), attributing to promoted selectivity (Fig. 2b). In the meanwhile, loading Co onto BiVO₄/Pd (i.e., CoOₓ/BiVO₄/Pd) improved water oxidation efficiency, resulting in a 3.9-fold enhancement of 20-min H₂O₂ generation (245 μM). Most importantly, with successful tuning of interfacial energetics, CoOₓ/BiVO₄/(Ag/Pd) enhanced the H₂O₂ generation by 2.1-fold (520 μM, Fig. 2a). Interestingly, when the

core/shell structure was reversed (i.e., Pd core and Ag shell, denoted as CoOₓ/BiVO₄/(Pd/Ag)), the photocatalytic H₂O₂ generation rate was much decreased, even lower than that of CoOₓ/BiVO₄/Pd (Fig. 2a). This result demonstrates that forming a BiVO₄/Ag junction at {010} reduction sites instead of random Ag loading is key to improve photocatalytic H₂O₂ generation.

The construction of BiVO₄/Ag junction did not impair the surface reaction selectivity. For instance, as compared to the low selectivity of BiVO₄/Ag towards H₂O₂ generation (13 %), the H₂O₂ generation selectivity of BiVO₄/(Ag/Pd) (81 %) was similar to that of BiVO₄/Pd (85 %, Fig. 2b), suggesting that the Pd shell fully covered Ag core and acted as the dominant H₂O₂ generation sites. H₂O₂ production on CoOₓ/BiVO₄/(Pd/Ag) proceeded via a two-electron oxygen-reduction path, as demonstrated by halted H₂O₂ production in the presence of IO₃⁻ as

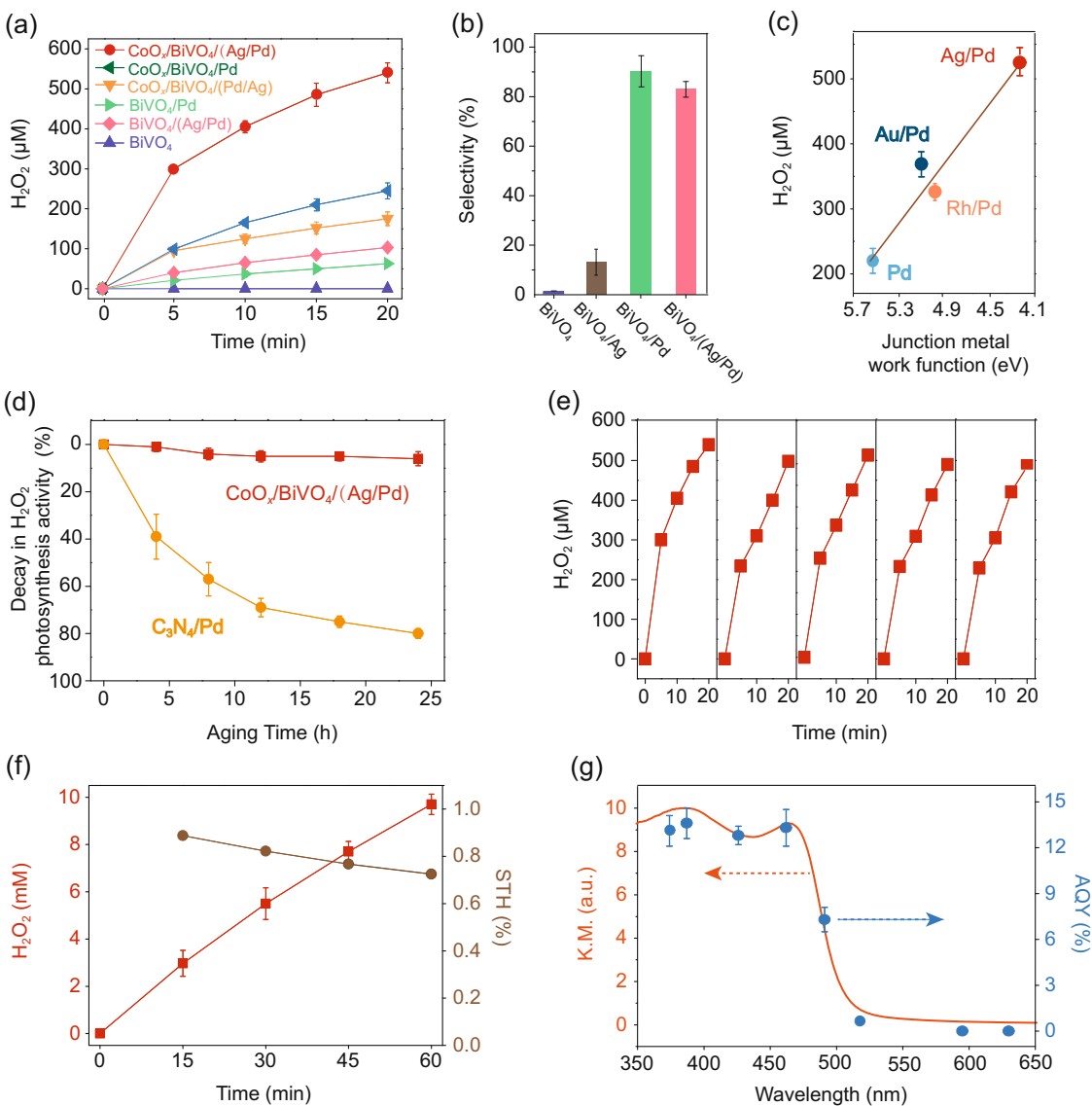

**Fig. 2 | Overall H$_2$O$_2$ photosynthesis activities. a** Time courses of photocatalytic H$_2$O$_2$ generation. Reaction conditions: photocatalyst, 1 mg/mL; 50 ml DI water saturated with O$_2$; light source, LED visible light, 300 mW cm$^{-2}$, $\lambda$ > 400 nm. **b** Selectivity of H$_2$O$_2$ production for BiVO$_4$, BiVO$_4$/Ag, BiVO$_4$/Pd, and BiVO$_4$/(Ag/Pd). Reaction conditions: photocatalyst, 1 mg/ml; 50 mL DI water with 0.1 M H$_3$BO$_3$ and 0.075 M ScCl$_3$ saturated with O$_2$ (pH 6.8), 10 v/v% methanol as electron donor; light source, LED visible light, 100 mW/cm$^2$, $\lambda$ > 400 nm. H$_2$O$_2$ selectivity is defined as the ratio of electrons utilized for H$_2$O$_2$ synthesis to the total number of electrons consumed (i.e., electrons donated by methanol). **c** Preparation of various core/shell cocatalyst and correlation between H$_2$O$_2$ photosynthesis performance and core junction metal work function. Reaction conditions: photocatalyst, 1 mg/ml; 50 mL DI water saturated with O$_2$ (pH 6.8); light source, LED visible light, 300 mW/cm$^2$, $\lambda$ > 400 nm; reaction time, 20 min. **d** Decay in H$_2$O$_2$ photosynthesis activity of CoO$_x$/BiVO$_4$/(Ag/Pd) and C$_3$N$_4$/Pd under •OH-rich conditions. Aging conditions:

photocatalyst, 1 mg/mL; 50 ml DI water with 25 mM H$_2$O$_2$; light source, 254 nm ultraviolet radiation light; •OH was generated via the reaction H$_2$O$_2$ + $hv$ → 2•OH. **e** Repetitive use of CoO$_x$/BiVO$_4$/(Ag/Pd) for H$_2$O$_2$ photosynthesis. Reaction conditions: photocatalyst amount, 1 mg/ml; 50 ml DI water saturated with O$_2$; light source, LED visible light, 300 mW cm$^{-2}$, $\lambda$ > 400 nm. **f** Time courses of photocatalytic H$_2$O$_2$ generation over CoO$_x$/Mo:BiVO$_4$/(Ag/Pd) and the corresponding STH efficiency. Reaction conditions: photocatalyst, 10 mg; photocatalyst, 1 mg/ml; 10 mL DI water with 0.1 M H$_3$BO$_3$ and 0.075 M ScCl$_3$ saturated with O$_2$ (pH 6.8); light source, xenon lamp solar simulator, 100 mW cm$^{-2}$, AM 1.5 G; irradiation area, 4.5 cm$^{-2}$. **g** Apparent quantum yield (AQY) of H$_2$O$_2$ photosynthesis over CoO$_x$/Mo:BiVO$_4$/(Ag/Pd) as a function of the incident light wavelength. Reaction conditions: photocatalyst, 10 mg; photocatalyst, 1 mg/ml; 10 mL DI water with 0.1 M H$_3$BO$_3$ and 0.075 M ScCl$_3$ saturated with O$_2$ (pH 6.8); light source, monochromatic LED light.

electron scavenger or under N$_2$-purged condition and enhanced H$_2$O$_2$ production when methanol as hole scavenger was added (Fig. S14).

To further show the effectiveness of tuning interfacial energetics on improving photosynthesis performance, we constructed Au/Pd and Rh/Pd core/shell structure cocatalysts and examined their H$_2$O$_2$ generation activity with BiVO$_4$ (Fig. 2c). The work functions of Au (5.1 eV) and Rh (5.0 eV) were higher than Ag (4.3 eV), yet lower than Pd (5.6 eV). Therefore, Au and Rh in theory should also be capable of lowering the Schottky barrier on {010} facets, yet to a lower extent than Ag.

The photocatalytic H$_2$O$_2$ generation shows a distinct negative correlation with metal work function (Fig. 2c), further demonstrating the validity of lowering Schottky barrier for enhanced charge separation and photosynthesis performance.

CoO$_x$/BiVO$_4$/(Ag/Pd) exhibits high chemical stability during H$_2$O$_2$ generation. After 24-h incubation in •OH-rich condition, CoO$_x$/BiVO$_4$/(Ag/Pd) displayed nominal decay in H$_2$O$_2$ production performance (Fig. 2d). In stark contrast, C$_3$N$_4$/Pd as a representative organic photocatalyst was highly susceptible to •OH, with ~80% decay in H$_2$O$_2$

production performance after 24-h incubation (Fig. 2d). Because photocatalytic $H_2O_2$ generation is inevitably accompanied by •OH generation ($H_2O_2 + h\nu \rightarrow 2$ •OH or $H_2O_2 + e^- + H^+ \rightarrow$ •OH $+ H_2O$), such resistance to •OH as shown by $CoO_x$/BiVO$_4$/(Ag/Pd) is pivotal for long-term $H_2O_2$ photosynthesis. The high stability of $CoO_x$/BiVO$_4$/(Ag/Pd) was further demonstrated by its stable catalytic performance over repetitive use up to five cycles (Fig. 2e). Notably, the photo-catalytic $H_2O_2$ generation over $CoO_x$/BiVO$_4$/(Ag/Pd) decayed over time in each cycle (Fig. 2e). This issue has been noticed in the pre-vious studies on photocatalytic $H_2O_2$ generation[12], which is caused by $H_2O_2$ self-decomposition instead of photocatalyst inactivation and can be solved by adding $H_2O_2$ stabilizers such as $Sc^{3+}$ (see data below)[34,35].

With exquisite tuning of interfacial energetics and surface reac-tions, BiVO$_4$ exhibits a high overall $H_2O_2$ photosynthesis efficiency. Here we applied Mo-doped BiVO$_4$ (Mo:BiVO$_4$) for further photo-catalytic performance evaluation because doping the $V^{5+}$ sites of BiVO$_4$ with $Mo^{6+}$ can increase its bulk conductivity[36,37]. Without any sacrificial reagent, $CoO_x$/Mo:BiVO$_4$/(Ag/Pd) produced 9.7 mM $H_2O_2$ in one hour, corresponding to an AQY of 3.0% and a STH of 0.73% at full spectrum (Fig. 2f). Such an efficiency leveraged the record for inorganic semi-conductors by 252% and is comparable with the efficiencies of most organic semiconductors (Table S1). This $H_2O_2$ production perfor-mance was confirmed to be reproducible with three batches of $CoO_x$/Mo:BiVO$_4$/(Ag/Pd) (Fig. S15). The wavelength-dependent AQYs mea-sured by light-emitting diode (LED) light irradiation agree well with the absorption spectrum (Fig. 2g), suggesting that $H_2O_2$ was generated following the bandgap excitation of BiVO$_4$. The AQY at 420 nm was determined to be 13.1%, the highest reported for inorganic semi-conductors to the best of our knowledge (Table S1).

## Time-resolved-spectroscopic analyses revealing enhanced charge separation through interfacial energetics tuning

To investigate the impact of surface energetics tuning on charge separation, charge-carrier dynamics in $CoO_x$/BiVO$_4$/(Ag/Pd) and $CoO_x$/BiVO$_4$/Pd were thoroughly studied by transient absorption spectro-scopy (TAS). The transient kinetics of photogenerated charge carriers in BiVO$_4$ were monitored upon band-gap excitation using 470 nm (~2.64 eV) laser pulses, with photogenerated surface trapped holes probed at 505 nm and photogenerated free/shallowly trapped elec-trons probed at 2000 nm[38–40].

As shown in Fig. 3a, $CoO_x$/BiVO$_4$/(Ag/Pd) exhibits a longer electron lifetime than $CoO_x$/BiVO$_4$/Pd, indicating an enhanced charge-separation process in the former case with a higher $\Delta V$. Yet, considering that $CoO_x$/BiVO$_4$/(Ag/Pd) was about three times more active than $CoO_x$/BiVO$_4$/Pd for $H_2O_2$ generation, their dif-ference on electron decay is smaller than expectation. We therefore infer that a higher $\Delta V$ not only enhances charge separation by preserving charge carriers from recombination, but also by driving these charge carriers to the corresponding coca-talysts (Fig. 3e). As a result, electron accumulation in BiVO$_4$ is competed by the rapid electron transfer to Ag/Pd.

To verify whether a higher $\Delta V$ in $CoO_x$/BiVO$_4$/(Ag/Pd) can enhance charge separation by driving charge carriers to cocatalysts, we further investigated the hole decay of $CoO_x$/BiVO$_4$/Pd and $CoO_x$/BiVO$_4$/(Ag/Pd). Since photogenerated holes exhibit comparably strong signal and long lifetime in the microsecond-millisecond timescales[39,40], we also show the hole kinetics of bare BiVO$_4$ and BiVO$_4$/Pd for a detailed comparison. As shown in Fig. 3b, BiVO$_4$/Pd exhibits a longer hole lifetime than bare BiVO$_4$ due to electron transfer to Pd. Loading $CoO_x$ on BiVO$_4$/Pd (i.e., $CoO_x$/BiVO$_4$/Pd) further prolonged the hole lifetime, even though the sluggish decay of long-lived holes was partially offset by hole transfer to $CoO_x$. We attribute the prolonged hole lifetime to (i) synergistically enhanced charge separation by Pd and $CoO_x$ coloading,

and (ii) more efficient electron transfer to Pd than hole transfer to $CoO_x$, which have been confirmed in our previous study[12].

$CoO_x$/BiVO$_4$/(Ag/Pd) shows a shorter hole lifetime than $CoO_x$/BiVO$_4$/Pd, proving that the elevated $\Delta V$ enhanced migration of holes from bulk BiVO$_4$ to $CoO_x$. Such enhanced hole migration can lead to higher hole accumulation in the $CoO_x$ of $CoO_x$/BiVO$_4$/(Ag/Pd) and accelerate surface reactions (Fig. 3e). As a proof, we examined the reactivity of holes in $CoO_x$ of $CoO_x$/BiVO$_4$/(Ag/Pd) and $CoO_x$/BiVO$_4$/Pd towards oxidation of formic acid (HCOOH), a representative hole scavenger[41,42]. For both samples, HCOOH as a hole scavenger acceler-ated the hole decays (Figs. 3c and 3d), where the acceleration was more significant in $CoO_x$/BiVO$_4$/(Ag/Pd) than in $CoO_x$/BiVO$_4$/Pd. For instance, at 1 ms, the hole signal intensity for $CoO_x$/BiVO$_4$/Pd decreased by 69%, while that for $CoO_x$/BiVO$_4$/(Ag/Pd) decreased by 87%. This indicates that 31% of holes remained in BiVO$_4$ for $CoO_x$/BiVO$_4$/Pd, whereas only 13% for $CoO_x$/BiVO$_4$/(Ag/Pd). The corre-sponding percentage decrease of the TA signal represents the popu-lation of holes utilized for HCOOH oxidation.

This is a notable result of elevated $\Delta V$ that enhanced the charge separation by driving higher extent of holes transferred to $CoO_x$ for HCOOH oxidation. Collectively, the TAS results indicate that lowering the Schottky barrier at {010} facets by constructing BiVO$_4$/Ag junction effectively enhances charge separation by surviving charge carriers as well as driving these charge carriers to corresponding cocatalysts for surface reactions (Fig. 3e).

## Electrical simulation revealing impacts of surface energetics on charge separation

To quantitatively study the effects of surface energetics on charge separation, COMSOL Multiphysics was applied to simulate the photocatalytic $H_2O_2$ generation reactions on $CoO_x$/BiVO$_4$/Pd and $CoO_x$/BiVO$_4$/(Ag/Pd) particles. A three-dimensional (3D) model for BiVO$_4$ particles (Fig. S16) was used in the simulation to accurately represent experimental conditions. The optoelectronic parameters used in the COMSOL simulation were obtained from our previous studies and literature as summarized in Table S2. The charge-carrier generation in the particle model was detailed in Section S1, Fig. S17–19. The experimental photoatalytic $H_2O_2$ generation rates were converted to photocurrent densities (Sec-tion S2) in order to quantitatively compare with the photocurrent densities obtained by comsol simulation.

We dissected the operating cocatalyst-loaded BiVO$_4$ as a combi-nation of a solar cell and an electrocatalyst cell (Section S3, Fig. S20)[43]. The solar cell was simulated by a 3D particle model with reduction (top/bottom) and oxidation (side) facets as the cathode and anode, respectively (Fig. 4a, b). The barrier height at the oxidation facets was fixed at 1.23 V, while that at the reduction facets was varied to obtain the corresponding light-responsive $J–V$ curves. The dark $J–V$ curve of the electrocatalyst, $H_2O_2$ generation on cathode and $O_2$ evolution on anode, was experimentally measured (Fig. S21). The intersect of the light-responsive $J–V$ curve from the simulated solar cell and $J–V$ curve from the electrocatalyst cell represents the photocurrent density of cocatalyst-loaded BiVO$_4$ particles (Fig. S22).

The photocurrent densities of cocatalyst-loaded BiVO$_4$ par-ticles with various barrier heights at the reduction facets were shown in Fig. 4c. The photocurrent densities for $CoO_x$/BiVO$_4$/Pd and $CoO_x$/BiVO$_4$/(Ag/Pd) were determined to be 0.0081 and 0.0257 mA cm$^{-2}$, corresponding to a barrier height at the reduc-tion facets of 0.32 and 0.09 V, respectively. This result indicates that Ag between BiVO$_4$ and Pd decreased the Schottky barrier at the reduction facets of BiVO$_4$ by 0.23 V. Notably, this value is larger than that (0.13 V) obtained by UPS (Fig. 1e), which is rea-sonable because the UPS results reflected the energetics of not only reduction facets but also the oxidation ones.

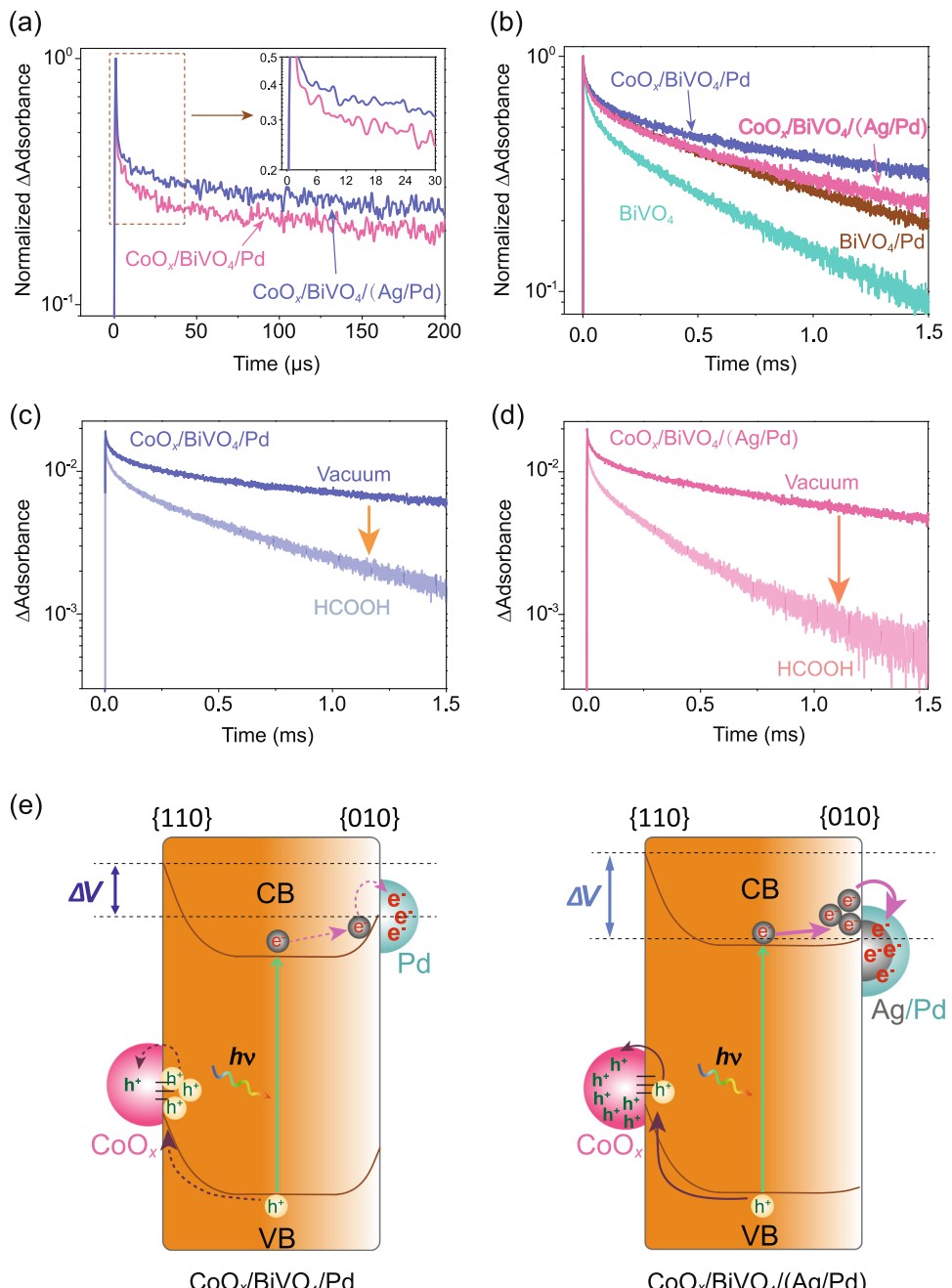

**Fig. 3 | Charge-carrier dynamics. a, b** Transient profiles of **a** free/shallowly trapped electrons probed at 2000 nm and **b** trapped holes probed at 505 nm. Photoexcitation of the samples was performed using 470 nm laser pulses (duration: 6 ns, fluence: 3 mJ/pulse, frequency: 1 Hz). Measurements were carried out in vacuum (base pressure: ~ $10^{-5}$ Torr). **c, d** The transient profiles of trapped holes probed at 505 nm for **c** $CoO_x/BiVO_4/Pd$ and **d** $CoO_x/BiVO_4/(Ag/Pd)$ in vacuum and in the presence of HCOOH vapor. Photoexcitation of the samples was performed using 470 nm laser pulses (duration: 6 ns, fluence: 3 mJ/pulse, frequency: 1 Hz). Measurements were carried out in vacuum or in the presence of 20 Torr HCOOH. **e** Schematic illustration of the charge-separation process enhanced by surface energetics tuning.

After extracting the barrier heights of $CoO_x/BiVO_4/Pd$ and $CoO_x/BiVO_4/(Ag/Pd)$ at their reduction facets, we further compared their optoelectronic properties including band energy diagram and mobile charge carrier density (Fig. 4d–m). With the barrier height at the cathodic site even slightly higher than that of the anodic site (Fig. 4d, j), the asymmetry in energetics of $CoO_x/BiVO_4/Pd$ is relatively small. This results in the back-flow and poor accumulation of charge carriers on the surface of $BiVO_4$ (Fig. 4f, h, l). In comparison, with Schottky barrier lowered by Ag, $CoO_x/BiVO_4/(Ag/Pd)$ particle exhibits a larger asymmetry in energetics (Fig. 4e, k), which efficiently drive electrons and

holes to the cathodic and anodic sites, respectively (Fig. 4g, i, m). Overall, the simulations on photocarrier dynamics in $BiVO_4$ particles demonstrate that constructing a $BiVO_4$/Ag junction effectively lowered the Schottky barrier at reduction sites, creating a high $\Delta V$ for improving charge separation and photosynthesis performance.

## Generality of interfacial-energetics-tuning strategy for enhancing artificial photosynthesis

Our surface-energetic-tuning strategy was further validated on $C_3N_4$ and $TiO_2$, two widely studied photocatalysts for artificial

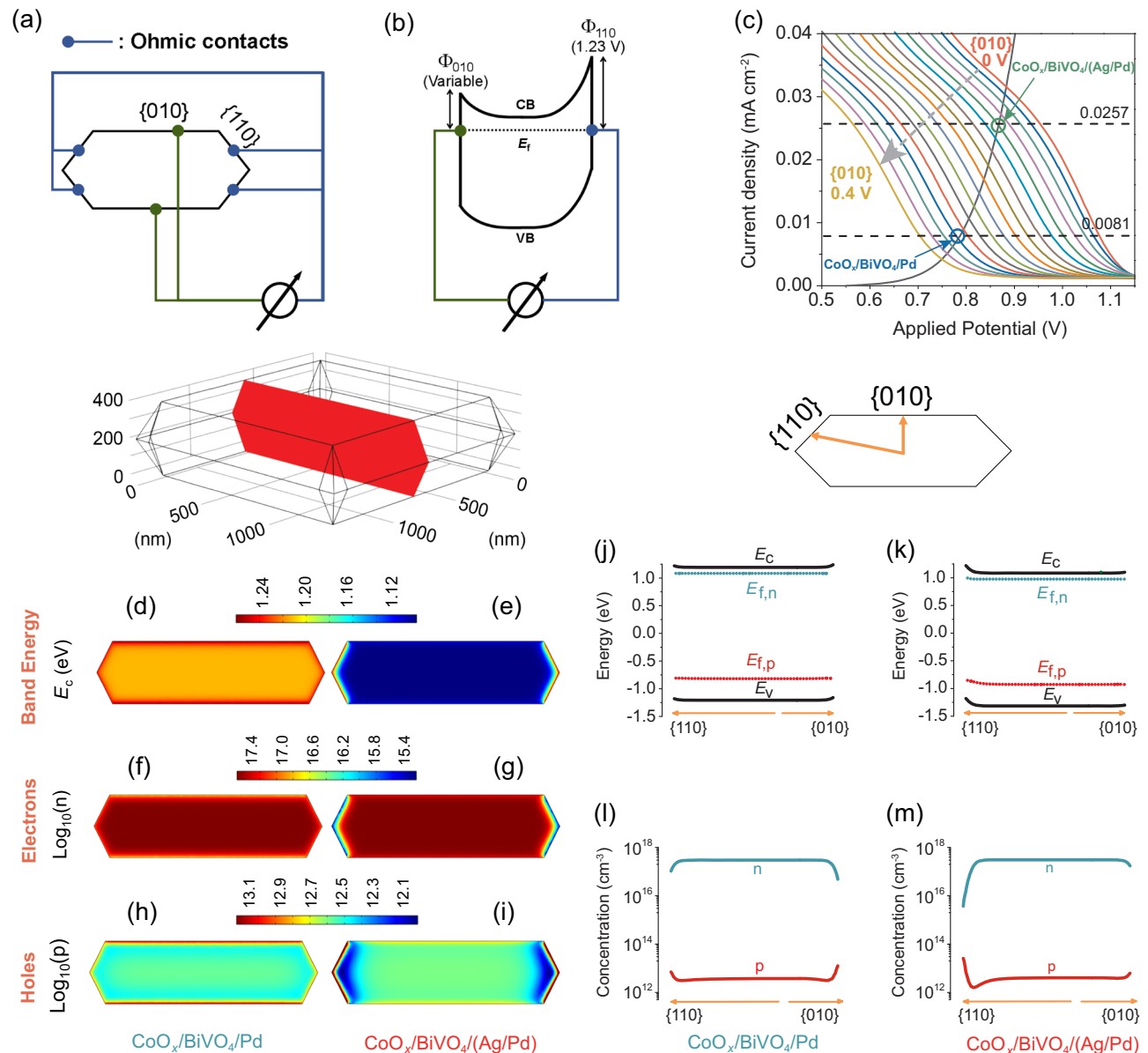

**Fig. 4 | Simulations of photocarrier distributions. a, b** Schematic model and band diagram of BiVO$_4$ as a solar cell. **c** Current density vs. applied potential with barrier heights at cathodic sites ($\Phi_{010}$) varied from 0 to 0.4 V (gray dotted arrow). The barrier height of the anodic side was fixed at 1.23 V. Two dash lines labeled with 0.0081 and 0.0257 mA cm$^{-2}$ were used to indicate the photocurrent densities converted from experimental H$_2$O$_2$ generation rates of CoO$_x$/BiVO$_4$/Pd and CoO$_x$/ BiVO$_4$/(Ag/Pd), respectively. The operating conditions of CoO$_x$/BiVO$_4$/Pd and CoO$_x$/BiVO$_4$/(Ag/Pd) were marked with two void circles. Their detailed optoelectronic properties are presented in the following figures. **d–i** 2D cross-sectional plots of the optoelectronic properties of a BiVO$_4$ particle, including conduction band energy (eV) (**d, e**), electron concentration (**f, g**), and hole concentration (**h, i**). **j–m** 1D plots of energy band diagram (**j, k**) and mobile charge carrier density (**l, m**).

photosynthesis (e.g, H$_2$, H$_2$O$_2$)[44,45]. C$_3$N$_4$/(Ag/Pd) and TiO$_2$/(Ag/Pd) was 1.7 and 1.4 times more active than C$_3$N$_4$/Pd and TiO$_2$/Pd on H$_2$O$_2$ photosynthesis, respectively (Figs. 5a and 5b). Such results clearly manifest the generality of our strategy on enhancing photocatalysis. Notably, the enhancement on C$_3$N$_4$ and TiO$_2$ is weaker than that on faceted BiVO$_4$. This is reasonable because without spatially separating the energetics on different facets, deposited Ag may disturb the energetics of the nearby oxidation sites and thus degrade the overall charge-separation process in photocatalysts.

Based on above results, we propose a general approach for effective interfacial-energetics-tuning (Fig. 5c): (i) apply facet engineering to spatially separate the surface energetics for electron and hole accumulation, (ii) selectively deposit metal/metal oxide species with desirable work functions on the required facets to tune the surface energetics for charge separation; and

(iii) selectively deposite catalysts on the metal/metal oxide species to tune surface reactions.

Our study validated the tuning of interfacial energetics as a general and effective approach to enhance charge separation, a key challenge in designing high-performance photocatalysts. The construction of core/shell Ag/Pd cocatalyst lowered the Schottky barrier at the {010} facets of BiVO$_4$ without impairing the surface reactions, leading to enhanced overall asymmetry in energetics and charge separation. With successful tuning of interfacial energetics, BiVO$_4$ generated H$_2$O$_2$ with an AQY of 3.0% and a STH efficiency of 0.73% at full spectrum, a new record for inorganic semiconductor-based systems. As first highlighted by BiVO$_4$ for H$_2$O$_2$ generation, such a strategy of tuning interfacial energetics can be generally applied towards other photosynthetic systems for promoting solar fuel production such as water splitting and CO$_2$ reduction.

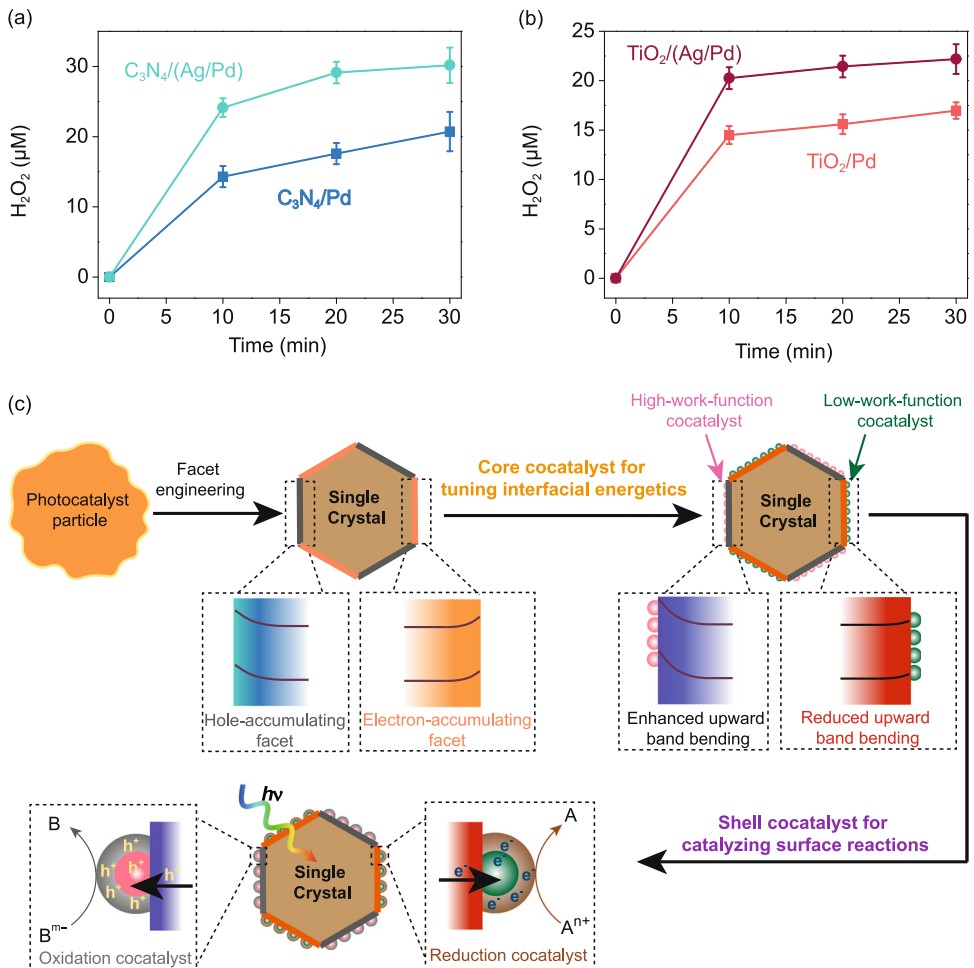

**Fig. 5 | Generality of interfacial-energetics-tuning strategy for enhancing artificial photosynthesis.** Time courses of photocatalytic $H_2O_2$ generation by **a** $C_3N_4$/Pd and $C_3N_4$/(Ag/Pd) and **b** $TiO_2$/Pd and $TiO_2$/(Ag/Pd). Reaction conditions: photocatalyst, 1 mg/mL; 50 ml DI water saturated with $O_2$; light source, LED visible light (300 mW cm$^{-2}$, $\lambda > 400$ nm) for $C_3N_4$ and UVA light ($\lambda = 365$ nm) for $TiO_2$. **c** A general approach for effective interfacial-energetics-tuning and enhanced artificial photosynthesis.

## Methods

### Catalyst preparation
Single crystal $BiVO_4$ was prepared by heating the mixture of $K_2CO_3$ (1.047 g) and $V_2O_5$ (2.272 g) in a ceramic crucible at a heating rate of 1.5 °C/min to 450 °C and annealing for 5 h in a muffle furnace. The obtained $K_3V_5O_{14}$ (2 g) was mixed with $Bi(NO_3)_3$•$5H_2O$ (0.326 g) and dispersed in 50 mL water under ultrasonication for 30 min. The mixture was heated at 70 °C for 10 h under ultrasonication, separated by centrifugation, washed with deionized water, and dried at 70 °C for 8 h.

### Cocatalyst deposition
As-prepared $BiVO_4$ (0.2 g) was dispersed in 100 ml water, followed by addition of 0.1 mol $NaIO_3$ and 0.35 mL $Co(NO_3)_2$ solution (1.5 g/L). The mixture was irradiated for 3 h using a xenon lamp solar simulator (model 300 DUV; Perfect Light, Inc., light intensity = 0.1 W cm$^{-2}$, $\lambda > 420$ nm), filtered, washed with deionized water, and dried at 60 °C for 8 h. As-prepared $CoO_x$/$BiVO_4$ (0.15 g) was dispersed in 100 ml pure water, followed by addition of 0.36 ml $AgNO_3$ solution (0.33 g/L). The mixture pH was adjusted to 8.5 and irradiated at $\lambda > 420$ nm for 20 h. As-prepared $CoO_x$/$BiVO_4$/Ag was dispersed in 100 ml pure water, followed by addition of 0.182 mL $Na_2PdCl_4$ solution (3.3 g/L). The mixture was irradiated at $\lambda > 420$ nm for 3 h. As-prepared $CoO_x$/$BiVO_4$/(Ag/Pd) was filtered, washed with deionized water, and dried at 60 °C for 8 h. Same photodeposition method was also applied in preparing $BiVO_4$/

Ag, $BiVO_4$/Pd, $CoO_x$/$BiVO_4$/Ag, $CoO_x$/$BiVO_4$/Pd, $CoO_x$/$BiVO_4$/(Rh/Pd), $CoO_x$/$BiVO_4$/(Au/Pd), and $CoO_x$/$BiVO_4$/(Pd/Ag) using respective cocatalyst precursors. Similarly, $C_3N_4$/Pd, $C_3N_4$/(Ag/Pd), $TiO_2$/Pd, and $TiO_2$/(Ag/Pd) were prepared.

### Photocatalyst characterizations
XPS measurements were performed with a Thermo Scientific 250Xi system with monochromatic Al Kα as the excitation source. XRD patterns were recorded with a Bruker D8 Advance X-ray diffractometer with Cu Kα radiation ($\lambda = 1.5406$ Å) operated at 40 kV and 40 mA. SEM images were taken using a Hitachi SU-8010 microscope equipped with EDS at 30 kV. TEM images were taken using a Hitachi 7650 microscope operated at 100 kV.

### Photocatalytic activity tests
Batch assessments of photocatalytic $H_2O_2$ production by different photocatalyst were conducted using a high throughput reactor (model slight; Perfect Light Inc., Fig. S23) equipped with LED visible light. The LED light spectrum is shown in Fig. S24. Photocatalyst (50 mg) was dispersed in 50 mL deionized water in a custom-made reactor by ultrasonication for 10 min and purged with $O_2$ for 20 min. $H_2O_2$ production was assessed under water bath ($12 \pm 0.5$ °C) with an input light intensity of 300 mW/cm$^2$ ($\lambda > 400$ nm; irradiation area = 7.1 cm$^2$). The photocatalytic $H_2O_2$ production performance was measured using a xenon lamp (model 300 DUV; Perfect Light Inc., Fig. S25). The

spectrum of xenon lamp and the standard AM1.5 G (ASTMG 173) are shown in Fig. S26. Photocatalyst (10 mg) was dispersed in 10 mL deionized water containing 0.1 M $H_3BO_3$ and 0.075 M $ScCl_3$ by ultrasonication for 10 min and purged with $O_2$ for 20 min. $H_2O_2$ production was assessed under water bath (12 ± 0.5 °C) with an input light intensity of 100 mW cm$^{-2}$ (AM 1.5 G; irradiation area = 4.5 cm$^2$).

At designated time points, 50 μL suspension was taken for analysis of $H_2O_2$ productions and diluted with phosphate buffer (pH=7.4) to a $H_2O_2$ concentration (1–15 μM) that is most suitable for accurate $H_2O_2$ quantification, followed by centrifugation. Afterwards, 50 μL supernatant was taken and mixed with 50 μL solutions containing phosphate buffer (50 mM, pH = 7.4), ampliflu red (100 μM), and horseradish peroxidase (0.05 U/mL). Ampliflu red selectively reacted with $H_2O_2$ in the presence of horseradish peroxidase and formed the product resorufin. Resorufin in the mixture solution was quantified using an Agilent high-performance liquid chromatography coupled to a photo-diode array detector (detection at 560 nm). The calibration curve in Fig. S27 was used to quantitatively analyze $H_2O_2$ production. HPLC analysis was carried out in a C18 column at 20 °C with an isocratic mobile phase of 55% sodium citrate buffer (with 10% methanol (v/v), pH 7.4) and 45% methanol (v/v) at a flow rate of 0.5 mL/min.

The photocatalytic $H_2O_2$ production selectivity was assessed using methanol as electron donor. $H_2O_2$ selectivity is defined as the ratio of electrons utilized for $H_2O_2$ synthesis to the total number of electrons consumed (i.e., electrons donated by methanol)[46]. The consumption amount of methanol was assessed by analyzing its oxidation product formaldehyde. Experimental suspensions contained 1 g/L photocatalyst and 10% methanol (v/v). The suspension was $O_2$ saturated by continuous purging during the irradiation. Nash's reagent was used for quantification of formaldehyde. Nash's reagent (0.5 mL) containing 2 M ammonia acetate, 30 mM acetyl acetone, and 35 mM acetic acid was mixed with the sample suspension at 1:1 ratio and heated in oven at 60 °C for 1 h. The absorption spectra of products were measured using UV-vis spectrometer at 415 nm. Formaldehyde standard solution was used to calibrate the absorption in relation to formaldehyde concentration.

### TAS measurements

Microsecond-millisecond TAS measurements were carried out using Nd:YAG lasers (Continuum, Surelite I) and custom-built spectrometers[14,47]. Briefly, the TAS signals for the photoinduced photocarrier were monitored after band-gap excitation using 470 nm laser pulses (duration: 6 ns, fluence: 3 mJ pulse$^{-1}$). To probe the photoinduced electrons, the IR beam from the $MoSi_2$ coil was irradiated and focused on the film sample. The transmitted IR beam entered a monochromatic grating spectrometer and was detected by a mercury cadmium telluride (MCT) detector (Kolmar). The photoexcited electrons was monitored at 2000 nm. To probe the photogenerated holes, the white continuum beam emitted by a halogen lamp was focused on the sample. The reflected beam from the sample entered the monochromatic grating spectrometer and was detected by Si photodetector. The photogenerated holes was monitored at 505 nm (~2.45 eV). The output electric signal was then amplified with an AC-coupled amplifier (Stanford Research Systems, SR560, 1 MHz) that can monitor responses from 1 microsecond to several milliseconds. Ten to one thousand responses were averaged to obtain the transient decay profile at the probe wavelength. The time resolution of the spectrometer was limited to 1 μs by the bandwidth of the amplifier. The experiments were carried out in vacuum (base pressure ~ 10$^{-5}$ Torr) and at room temperature. For the measurement with formic acid, the pressure inside the reaction cell was at 20 Torr. For sample preparation, photocatalyst was

dispersed in isopropanol, drop-casted on a $CaF_2$ substrate, and dried in air to obtain a powder film with a density of ~ 1.3 mg/cm$^2$.

### Numerical simulation

Charge carrier dynamics in $BiVO_4$ was simulated using COMSOL Multiphysics. Detailed information on the charge-carrier distribution, and performance evaluation were elaborated in Section S1–S3.

## Data availability

Source data are provided with this paper. The data that support the findings of this study are available from the corresponding author upon reasonable request. Source data are provided with this paper.

## Code availability

Source data are provided with this paper. The code used for the electrical simulations are available from the corresponding author upon reasonable request.

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

## Acknowledgements

This work was supported by National Natural Science Foundation of China (NSFC, No. 22136004, 22006129) and JSPS KAKENHI (No. JP20K22556). We are grateful to Sudan Shen (State Key Laboratory of Chemical Engineering at Zhejiang University) and analysis center of agrobiology and environmental sciences for help in STEM and SEM measurements, respectively. Shu Hu and Rito Yanagi are supported by a grant from the Thistledown Foundation through the Research Corporation for Science Advancement Negative Emissions Science program.

## Author contributions

C.C. and Z.P. designed research. T.L., Z.P., and X.Z. synthesized the catalysts. T.L. and X.Z. conducted photocatalytic performance tests. T.L. conducted material characterization; Ko.K., J.J.M.V., and A.Y. performed TAS measurements and analysis; Y.R., Z.P., and S.H. conducted numerical simulation. T.L., C.C., Z.P., W.Y., B.C., and Ke.K. analyzed data; T.L., C.C., and Z.P. wrote the paper. All authors discussed the results and commented on the manuscript.

## Competing interests

The authors declare no competing interest.
