## [Peer Review File · Nature Communications]

A general interfacial-energetics-tuning strategy for enhanced artificial photosynthesisEditorial Note: Parts of this Peer Review File have been redacted as indicated to remove third-party material where no permission to publish could be obtained.

REVIEWER COMMENTS

Reviewer #1 (Remarks to the Author):

This paper is not easy to read, nor fully convincing. Furthermore many details are not easy to verify (for instance, refs. 16-20 are incomplete, figures in the main text can only be understood after having read the Methods reported in the final part of the manuscript, important experimental details are missing).

Reviewer #2 (Remarks to the Author):

The paper "A general interfacial-energetics-tuning strategy for enhanced artificial photosynthesis" by Liu et al. discussed a general strategy for modification of heterogeneous catalysts to enhance charge separation as well as photochemical performance. The authors modified inorganic catalyst BiVO₄ by incorporation of cobalt oxide and Ag/Pd core/shell cocatalyst, and the designed photocatalyst has shown good performance on H₂O₂ photosynthesis. The authors assumed cobalt oxide and Ag/Pd cocatalysts deposited on oxidation site and reduction site could enhance both water oxidation and oxygen reduction, respectively. It is noted that the unique Ag/Pd core/shell cocatalyst was well designed, which not only promoted the charge separation by lowering Schottky barrier due to the Ag core, but also maintain the surface kinetics and H₂O₂ generation selectivity due to the Pd shell. The structure of catalyst was well verified by characterization and electrical simulations. Based on logical design, the rate of photocatalytic H₂O₂ formation reached ca. 9.7 mM/h from water, which is impressive.

Overall, I think the authors follow an elegant approach in this paper by the modification of BiVO₄ photocatalyst. I believe this article is well written and has been analyzed in detailed. For these reasons, I would gladly support publication of the manuscript after the following concerns are adequately addressed.

1. I would recommend to better document the precedents related to the photocatalytic H₂O₂ generation. In this regard, I suggest to inspect this review article: *Angew.Chem.Int. Ed.* 2020, 59, 17356–17376.
2. In line 130, "were determined to be 2.22...", the values are not consistent with the values in Figure 1e.
3. In line 146, you mentioned that the loading amount of Pd was determined to be 0.3 wt% after optimization. But in Figure S11, 0.3 wt% did not give the best result. Please provide the reason that you chose 0.3 wt% Pd.
4. Some important literatures are missing in the reference: *J. Am. Chem. Soc.* 2022, 144, 22, 9902–9909; *J. Am. Chem. Soc.* 2022, 144, 2603; *ACS Catal.* 2021, 11, 14087; *Nat. Catal.* 2021, 4, 374–384; *Nat. Commun.* 2021, 12, 483; *Nat. Commun.* 2016, 7, 11470; *J. Am. Chem. Soc.* 2020, 142, 8641–8648.

Reviewer #3 (Remarks to the Author):

Similar articles have been shown in the recent literature and are not cited. The authors are suggested to include the following articles:

1. Photocatalytic H₂O₂ production from O₂ under visible light irradiation over phosphate ion-coated Pd nanoparticles-supported BiVO₄ K. Fuku et al., *Appl. Catal. B* 2020, 272, 119003
2. Bifunctional Pd-Ox Center at the Liquid-Solid-Gas Triphase Interface for H₂O₂ Photosynthesis M. Sun et al., *ACS Catal.* 2022, 12, 2138
3. Optimizing Oxygen and Intermediate HOO* Adsorption of Cu–Pd Alloy Cocatalyst for Boosting Photocatalytic H₂O₂ Production of BiVO₄ K. Wang et al., *Adv. Sustain. Systems* 2022, 6, 2200144

4. Mass-transfer control for selective deposition of well-dispersed AuPd cocatalysts to boost photocatalytic H₂O₂ production of BiVO₄ H. Shi et al., Chem. Engin. J. 2022, 443, 136429
The authors should make clear what is the novel aspect they bring in this submission. In addition, their recent publication (see below) should be cited directly. Actually, the authors refer to this in REF10 as well as Ref1 of Suppl. Information.

Overall photosynthesis of H₂O₂ by an inorganic semiconductor T. Liu et al., Nat. Commun. 2022, 13, 1034.

As I understand, the only difference of the current submission is that Ag nanoparticles are plus added. Furthermore, there is a recent review article which should be cited. Contains a useful Table with data about the yield of H₂O₂ evolution. The authors are suggested to include the best two examples of the Review Table in their Table S1 and add some comment about the comparison of yields (using the same units). The suggested review article is the following:

Photocatalytic Evolution of Hydrogen Peroxide: A Minireview N. Karamoschos et al., Energies 2022, 15, 6202.

The authors need to make clear the following:

In Figure S5 Which is the third component?

There are two components in Figure S7, please assign the second one.

The authors claim about selective positioning of metal nanoparticles in specific crystallographic planes. I think this statement is rather exaggerating and superficial. EDS images (Fig. 1c,d) show that at the top view we observe islands of metal assemblies, whereas in the side view we get rather dispersed metal nanoparticles. This is more clear in the case of palladium. Thus, the discussion about selective positioning is not standing at all and should be somehow omitted.

The authors are suggested to include EDS spectra of Co/Pd and Co/Ag in both views. In addition, please record Raman spectra in the top plane if possible and include in the manuscript. I think that the corresponding cobalt oxide peaks will be visible.

Omit the sentence: "SEM image shows that CoO_x particles

90 are uniformly distributed across the {110} facets of BiVO₄ (Fig. S4a)." There are also similar sentences with analogous meaning which should be omitted. Please perform an additional scavenging experiment (for example, using an hydroxy radical scavenger) in order to assess better the mechanistic path for H₂O₂ evolution. Thus, I do not recommend publication of the submission in the current version.

Response to Comments

A general interfacial-energetics-tuning strategy for enhanced artificial photosynthesis

Tian Liu, Zhenhua Pan, Kosaku Kato, Junie Jhon M. Vequizo, Rito Yanagi, Xiaoshan Zheng, Weilai Yu, Akira Yamakata, Baoliang Chen, Shu Hu, Kenji Katayama, Chiheng Chu

Reviewer 1

Comment 1.1 This paper is not easy to read, nor fully convincing. Furthermore many details are not easy to verify (for instance, refs. 16-20 are incomplete, figures in the main text can only be understood after having read the Methods reported in the final part of the manuscript, important experimental details are missing).

Response This work is conducted to solve a very fundamental question in artificial photosynthesis: How to reduce the Schottky barrier of a photocatalyst? This question is very important for enhancing artificial photosynthesis because the Schottky barrier at reduction sites is a major factor limiting the charge-separation process in a photocatalyst.

To answer this question, we combined the knowledge of semiconductor physics, catalysis chemistry, and material science, and developed a general interfacial-energetics-tuning strategy involving the application of a core/shell type cocatalyst. Such strategy is validated on various photocatalytic systems including BiVO₄, C₃N₄, and TiO₂ for H₂O₂ generation. We further applied time-resolved spectroscopy and numerical simulations to demonstrate that the application of a core/shell type cocatalyst can indeed tune the interfacial energetics as expected. We have revised the manuscript to further enhance the clarity, for instance:

Line 40: ‘Among primary photosynthetic systems, the particulate photocatalyst is considered as the most cost-effective one because of its high simplicity and scalability, yet its energy conversion efficiency still needs improving for viable applications.^{6,7}

In pursuit of high-performance particulate photocatalysts, a key challenge is to improve charge separation.⁸ Enhancing this process demands exquisitely tuning interfacial energetics between reduction and oxidation sites, and in the meantime not impairing the kinetics and selectivity for surface reactions.’

Line 52: ‘Such detrimental effects of a high Schottky barrier at the reduction sites have been verified in the previous studies (e.g., Pd-loaded BiVO₄ for H₂O₂ generation,¹² and Pt-loaded GaN or LaTiO₂N for H₂ evolution^{13,14}). *Therefore, the Schottky barrier at reduction sites is a major hurdle limiting charge separation and further artificial photosynthesis performance. Yet, how to lower the Schottky barrier at reduction sites remains a critical challenge for particulate photocatalysts (Fig. S1b).*’

Line 68: ‘Here, we report the interfacial energetics tuning in a particulate photocatalyst for photosynthesis of H₂O₂, an emerging liquid solar fuel substitute for gaseous H₂.¹⁷ *Organic semiconductors have been developed for photocatalytic H₂O₂ generation with reported STH up to 1.5 %.¹⁸⁻²¹ Yet, organic materials have a potential stability concern because*

photocatalytic H₂O₂ generation is inevitably accompanied by hydroxyl radical (\bullet OH) generation ($\text{H}_2\text{O}_2 + h\nu \rightarrow 2\bullet\text{OH}$ or $\text{H}_2\text{O}_2 + e^- + \text{H}^+ \rightarrow \bullet\text{OH} + \text{H}_2\text{O}$), which can damage organic structures.^{22,23} Our group recently developed an inorganic BiVO₄ photosynthesis system with a STH of 0.29 %. The AQY of the system is 5.8 % at 420 nm, indicating that majority of the charge carriers recombined during reaction and charge separation needs further tuned for higher photocatalytic performance.²⁴ In this study, we constructed a Ag/Pd binary cocatalyst with core/shell structure on the {010} reduction facets of BiVO₄.

Line 90: **‘Interfacial energetics tuning through applying core/shell type cocatalyst.** We first synthesized faceted BiVO₄ particles using a solid-liquid-reaction method.’

Line 131: **‘Fig. 1. Facet-selective loading of cocatalysts on BiVO₄ and interfacial energetics tuning with Ag/Pd core/shell cocatalyst.’**

Line 156: **‘Enhanced photocatalytic H₂O₂ generation with interfacial energetics tuning.** We first compare the photocatalytic H₂O₂ generation performance of different particulate photocatalysts under visible light irradiation ($\lambda > 400$ nm) in pure water.’

Line 230: **‘Time-resolved-spectroscopic analyses revealing enhanced charge separation through interfacial energetics tuning.** To investigate the impact of surface energetics tuning on charge separation, charge-carrier dynamics in CoO_x/BiVO₄/(Ag/Pd) and CoO_x/BiVO₄/Pd were thoroughly studied by transient absorption spectroscopy (TAS).’

Line 278: **‘Electrical simulation revealing impacts of surface energetics on charge separation.** To quantitatively study the effects of surface energetics on charge separation, COMSOL Multiphysics was applied to simulate the photocatalytic H₂O₂ generation reactions on CoO_x/BiVO₄/Pd and CoO_x/BiVO₄/(Ag/Pd) particles.’

Line 325: **‘Generality of interfacial-energetics-tuning strategy for enhancing artificial photosynthesis.** Our surface-energetic-tuning strategy was further validated on C₃N₄ and TiO₂, two widely studied photocatalysts for artificial photosynthesis (e.g, H₂, H₂O₂).’

Line 339: **‘Fig. 5. Generality of interfacial-energetics-tuning strategy for enhancing artificial photosynthesis.** Time courses of photocatalytic H₂O₂ generation by (a) C₃N₄/Pd and C₃N₄/(Ag/Pd) and (b) TiO₂/Pd and TiO₂/(Ag/Pd).’

We thank the reviewer for reminding us of the format of references. We have revised the references accordingly and further checked the reference list in our manuscript:

Line 502: ‘[24] Liu, T. *et al.* Overall photosynthesis of H₂O₂ by an inorganic semiconductor. *Nat. Commun.* **13**, 1034 (2022).’

Line 503: ‘[25] Zhao, W. *et al.* Accelerated synthesis and discovery of covalent organic framework photocatalysts for hydrogen peroxide production. *J. Am. Chem. Soc.* **144**, 9902-9909 (2022).’

Line 505: ‘[26] Tian, Q. *et al.* Nanospatial charge modulation of monodispersed polymeric microsphere photocatalysts for exceptional hydrogen peroxide production. *Small* **17**, 2103224 (2021).’

Line 515: ‘[31] Li, R. G. *et al.* Spatial separation of photogenerated electrons and holes among {010} and {110} crystal facets of BiVO₄. *Nat. Commun.* **4**, 1432 (2013).’

We have also added more experimental details in the figure caption to enhance the clarity:

Line 137: '(f) Schematic illustration of {010} reduction facet interfacial energetics tuning through Ag/Pd core/shell cocatalyst construction on BiVO₄.'

Line 213: 'Selectivity of H₂O₂ production for BiVO₄, BiVO₄/Ag, BiVO₄/Pd, and BiVO₄/(Ag/Pd). Reaction conditions: photocatalyst, 1 mg/ml; 50 mL DI water with 0.1 M H₃BO₃ and 0.075 M ScCl₃ saturated with O₂ (pH 6.8), 10 v/v% methanol as electron donor; light source, LED visible light, 100 mW/cm², $\lambda > 400$ nm. H₂O₂ selectivity is defined as the ratio of electrons utilized for H₂O₂ synthesis to the total number of electrons consumed (i.e., electrons donated by methanol).'

Line 215: 'Preparation of various core/shell cocatalyst and correlation between H₂O₂ photosynthesis performance and core junction metal work function.'

Line 276: '(e) Schematic illustration of the charge-separation process enhanced by surface energetics tuning.'

Reviewer 2

Comment 2.1 *A general interfacial-energetics-tuning strategy for enhanced artificial photosynthesis” by Liu et al. discussed a general strategy for modification of heterogeneous catalysts to enhance charge separation as well as photochemical performance. The authors modified inorganic catalyst BiVO₄ by incorporation of cobalt oxide and Ag/Pd core/shell cocatalyst, and the designed photocatalyst has shown good performance on H₂O₂ photosynthesis. The authors assumed cobalt oxide and Ag/Pd cocatalysts deposited on oxidation site and reduction site could enhance both water oxidation and oxygen reduction, respectively. It is noted that the unique Ag/Pd core/shell cocatalyst was well designed, which not only promoted the charge separation by lowering Schottky barrier due to the Ag core, but also maintain the surface kinetics and H₂O₂ generation selectivity due to the Pd shell. The structure of catalyst was well verified by characterization and electrical simulations. Based on logical design, the rate of photocatalytic H₂O₂ formation reached ca. 9.7 mM/h from water, which is impressive. Overall, I think the authors follow an elegant approach in this paper by the modification of BiVO₄ photocatalyst. I believe this article is well written and has been analyzed in detailed. For these reasons, I would gladly support publication of the manuscript after the following concerns are adequately addressed.*

Response We thank the reviewer for the efforts on improving our manuscript as well as the encouraging comments.

Comment 2.2 *I would recommend to better document the precedents related to the photocatalytic H₂O₂ generation. In this regard, I suggest to inspect this review article: Angew.Chem.Int. Ed.2020,59,17356–17376.*

Response Based on the reviewer’s suggestion, we have added more information on the precedents related to the photocatalytic H₂O₂ generation:

Line 68: ‘Here, we report the interfacial energetics tuning in a particulate photocatalyst for photosynthesis of H₂O₂, an emerging liquid solar fuel substitute for gaseous H₂.¹⁷ Organic semiconductors have been developed for photocatalytic H₂O₂ generation with reported STH up to 1.5 %.¹⁸⁻²¹ Yet, organic materials have a potential stability concern because photocatalytic H₂O₂ generation is inevitably accompanied by hydroxyl radical (•OH) generation (H₂O₂ + hν → 2•OH or H₂O₂ + e⁻ + H⁺ → •OH + H₂O), which can damage organic structures.^{22,23} Our group recently developed an inorganic BiVO₄ photosynthesis system with a STH of 0.29 %. The AQY of the system is 5.8 % at 420 nm, indicating that majority of the charge carriers recombined during reaction and charge separation needs further tuned for higher photocatalytic performance.²⁴ In this study, we constructed a Ag/Pd binary cocatalyst with core/shell structure on the {010} reduction facets of BiVO₄.’

Line 486: ‘[17] Hou, H. L., Zeng, X. K. & Zhang, X. W. Production of hydrogen peroxide by photocatalytic processes. *Angew. Chem. Int. Edit.* **59**, 17356-17376 (2020).’

Line 488: ‘[18] Kofuji, Y. *et al.* Carbon nitride-aromatic diimide-graphene nanohybrids: metal-free photocatalysts for solar-to-hydrogen peroxide energy conversion with 0.2% efficiency. *J. Am. Chem. Soc.* **138**, 10019-10025 (2016).’

Line 490: ‘[19] Isaka, Y. *et al.* Photocatalytic production of hydrogen peroxide through

selective two-electron reduction of dioxygen utilizing amine-functionalized MIL-125 deposited with nickel oxide nanoparticles. *Chem. Commun.* **54**, 9270-9273 (2018).'

Line 493: '[20] Shiraishi, Y. *et al.* Resorcinol-formaldehyde resins as metal-free semiconductor photocatalysts for solar-to-hydrogen peroxide energy conversion. *Nat. Mater.* **18**, 985-993 (2019).'

Comment 2.3 In line 130, "were determined to be 2.22...", the values are not consistent with the values in Figure 1e.

Response We have revised the manuscript accordingly:

Line 142: 'The (E_v-E_f) of BiVO₄, BiVO₄/Ag, BiVO₄/Pd and BiVO₄/(Ag/Pd) were determined to be 2.22, 2.35, 2.25, 2.36 eV, respectively (Fig. 1e).'

Comment 2.4 In line 146, you mentioned that the loading amount of Pd was determined to be 0.3 wt% after optimization. But in Figure S11, 0.3 wt% did not give the best result. Please provide the reason that you chose 0.3 wt% Pd.

Response We thank the reviewer for pointing out this mistake. We have revised the sentence accordingly:

Line 159: 'The performance of CoO_x/BiVO₄/(Ag/Pd) was optimized by adjusting the loading amount of Co, Ag, and Pd to 0.3 wt%, 0.08 wt%, 0.4 wt%, respectively (Fig. S11-S13).'

Comment 2.5 Some important literatures are missing in the reference: *J. Am. Chem. Soc.* 2022, 144, 22, 9902–9909; *J. Am. Chem. Soc.* 2022, 144, 2603; *ACS Catal.* 2021, 11, 14087; *Nat. Catal.* 2021, 4, 374–384; *Nat. Commun.* 2021, 12, 483; *Nat. Commun.* 2016, 7, 11470; *J. Am. Chem. Soc.* 2020, 142, 8641–8648.

Response We have added these important literatures on H₂O₂ photosynthesis in the reference:

Line 78: 'BiVO₄ is a visible-light-responsive photocatalyst promising for long-term H₂O₂ generation because it is resistive to •OH (a potent oxidant inevitably formed during H₂O₂ generation) compared to the organic counterparts.^{25-27'}

Line 503: '[25] Zhao, W. *et al.* Accelerated synthesis and discovery of covalent organic framework photocatalysts for hydrogen peroxide production. *J. Am. Chem. Soc.* **144**, 9902-9909 (2022).'

Line 39: 'Among primary photosynthetic systems, the particulate photocatalyst is considered as the most cost-effective one because of its high simplicity and scalability, yet its energy conversion efficiency still needs improving for viable applications.^{6,7'}

Line 462: '[6] Mase, K., Yoneda, M., Yamada, Y. & Fukuzumi, S. Seawater usable for production and consumption of hydrogen peroxide as a solar fuel. *Nat. Commun.* **7**, 11470 (2016).'

Line 464: '[7] Gopakumar, A. *et al.* Lignin-supported heterogeneous photocatalyst for the direct generation of H₂O₂ from seawater. *J. Am. Chem. Soc.* **144**, 2603-2613 (2022).'

Line 84: 'Without using any sacrificial agent, the rationally designed BiVO₄ photocatalyst achieved a new record among inorganic photocatalysts (Table S1).'

RF523	O ₂	420-700 nm	333K	2067	8	H ₂ O	-	0.5	8
Sb-SAPC	O ₂	AM1.5	298K	588	17.6	H ₂ O	-	0.61	9
Co ₁ /AQ/C ₃ N ₄	O ₂	AM 1.5	Room temperatu	62	-	H ₂ O	0.054	0.014	10

Reference in **Supplementary Information**: '[9] Teng, Z. Y. *et al.* Atomically dispersed antimony on carbon nitride for the artificial photosynthesis of hydrogen peroxide. *Nat. Catal.* **4**, 374-384 (2021).'

Reviewer 3

Comment 3.1 *Similar articles have been shown in the recent literature and are not cited. The authors are suggested to include the following articles:*

1. Photocatalytic H₂O₂ production from O₂ under visible light irradiation over phosphate ion-coated Pd nanoparticles-supported BiVO₄ K. Fuku et al., *Appl. Catal. B* 2020, 272, 119003

2. Bifunctional Pd-Ox Center at the Liquid-Solid-Gas Triphase Interface for H₂O₂ Photosynthesis M. Sun et al., *ACS Catal.* 2022, 12, 2138

3. Optimizing Oxygen and Intermediate HOO* Adsorption of Cu-Pd Alloy Cocatalyst for Boosting Photocatalytic H₂O₂ Production of BiVO₄ K. Wang et al., *Adv. Sustain. Systems* 2022, 6, 2200144

4. Mass-transfer control for selective deposition of well-dispersed AuPd cocatalysts to boost photocatalytic H₂O₂ production of BiVO₄ H. Shi et al., *Chem. Engin. J.* 2022, 443, 136429.

Response We have added these important literatures on H₂O₂ photosynthesis in the reference:

Line 78: ‘BiVO₄ is a visible-light-responsive photocatalyst promising for long-term H₂O₂ generation because it is resistive to •OH (a potent oxidant inevitably formed during H₂O₂ generation) compared to the organic counterparts.²⁵⁻²⁷ The Pd shell served to steer the oxygen-reduction pathway to two-electron process for H₂O₂ synthesis²⁸⁻³⁰, while the Ag core served to lower the Schottky barrier between BiVO₄ and Pd.’

Line 507: ‘[27] Sun, M. *et al.* Bifunctional Pd-Ox center at the liquid–solid–gas triphase interface for H₂O₂ photosynthesis. *ACS Catal.* 12, 2138-2149 (2022).’

Line 509: ‘[28] Fuku, K. *et al.* Photocatalytic H₂O₂ production from O₂ under visible light irradiation over phosphate ion-coated Pd nanoparticles-supported BiVO₄. *Appl. Catal. B-Environ.* 272, 119003 (2020).’

Line 511: ‘[29] Shi, H. Y. *et al.* Mass-transfer control for selective deposition of well-dispersed AuPd cocatalysts to boost photocatalytic H₂O₂ production of BiVO₄. *Chem. Eng. J.* 443, 136429 (2022).’

Line 513: ‘[30] Wang, K. Y. *et al.* Optimizing Oxygen and Intermediate HOO* Adsorption of Cu-Pd Alloy Cocatalyst for Boosting Photocatalytic H₂O₂ Production of BiVO₄. *Adv. Sustain. Syst.* 6, 2200144 (2022).’

Comment 3.2 *The authors should make clear what is the novel aspect they bring in this submission. In addition, their recent publication (see below) should be cited directly. Actually, the authors refer to this in Ref10 as well as Ref1 of Suppl. Information. Overall photosynthesis of H₂O₂ by an inorganic semiconductor T. Liu et al., *Nat. Commun.* 2022, 13, 1034. As I understand, the only difference of the current submission is that Ag nanoparticles are plus added*

Response We thank the reviewer for the opportunity to clarify the novelty of this work. This work is conducted based on a very fundamental question in artificial photosynthesis: How to reduce the Schottky barrier at reduction site of a photocatalyst? This question is very

important for improving the efficiency of artificial photosynthesis because the Schottky barrier is a major factor limiting the charge separation in a photocatalyst.

To answer this question, we developed a general interfacial-energetics-tuning strategy involving the application of a core/shell type cocatalyst. Such strategy is validated on various photocatalytic systems (BiVO₄, TiO₂, and C₃N₄) and proven efficient in improving the overall artificial photosynthesis efficiency of H₂O₂. We further applied time-resolved spectroscopy and numerical simulations to demonstrate that the application of a core/shell type cocatalyst can indeed enhanced the asymmetric interfacial energetics as expected.

We would like to note that the focus and novelty of this work is totally different from our recent work. Our recent publication (*Nat. Commun.* **13**, 1034 (2022)) focused on developing an efficient and stable inorganic system for H₂O₂ generation. In contrast, this work is to develop a general interfacial-energetics-tuning strategy to enhance charge separation. This strategy is not only applied to enhance the well-developed BiVO₄ system reported in our previous publication but also validated on other widely-studied photocatalysts including TiO₂ and C₃N₄. We also suspect that this strategy is applicable to particulate systems for water splitting and CO₂ reduction, as mentioned in our conclusion ‘As first highlighted by BiVO₄ for H₂O₂ generation, such a strategy of tuning interfacial energetics can be generally applied towards other photosynthetic systems for promoting solar fuel production such as water splitting and CO₂ reduction.’ We therefore believe that this study will be impactful in the field of particulate photosynthesis.

We have revised the manuscript to further enhance the clarity, for instance:

Line 39: ‘Among primary photosynthetic systems, the particulate photocatalyst is considered as the most cost-effective one because of its high simplicity and scalability, yet its energy conversion efficiency still needs improving for viable applications.’^{6,7}

In pursuit of high-performance particulate photocatalysts, a key challenge is to improve charge separation.⁸ Enhancing charge separation demands exquisitely tuning interfacial energetics between reduction and oxidation sites, and in the meantime not impairing the kinetics and selectivity for surface reactions.’

Line 462: ‘[6] Mase, K., Yoneda, M., Yamada, Y. & Fukuzumi, S. Seawater usable for production and consumption of hydrogen peroxide as a solar fuel. *Nat. Commun.* **7**, 11470 (2016).’

Line 464: ‘[7] Gopakumar, A. *et al.* Lignin-supported heterogeneous photocatalyst for the direct generation of H₂O₂ from seawater. *J. Am. Chem. Soc.* **144**, 2603-2613 (2022).’

Line 466: ‘[8] Yanagi, R., Zhao, T. S., Solanki, D., Pan, Z. H. & Hu, S. Charge separation in photocatalysts: mechanisms, physical parameters, and design principles. *ACS Energy Lett.* **7**, 432-452 (2022).’

Line 52: ‘Such detrimental effects of a high Schottky barrier at the reduction sites have been verified in the previous studies (e.g., Pd-loaded BiVO₄ for H₂O₂ generation¹², and Pt-loaded GaN or LaTiO₂N for H₂ evolution^{13,14}). Therefore, the Schottky barrier at reduction sites is a major hurdle limiting charge separation and further artificial photosynthesis performance. Yet, how to lower the Schottky barrier at reduction sites remains a critical challenge for particulate photocatalysts (Fig. S1b).’

Comment 3.3 Furthermore, there is a recent review article which should be cited. Contains a useful Table with data about the yield of H₂O₂ evolution. The authors are suggested to include the best two examples of the Review Table in their Table S1 and add some comment about the comparison of yields (using the same units). The suggested review article is the following: Photocatalytic Evolution of Hydrogen Peroxide: A Minireview N. Karamoschos et al., Energies 2022, 15, 6202.

Response We thank the reviewer for reminding us of such an important review article. We have added it to the reference list as reference [5] to support our discussion. Furthermore, we added the suggested examples (red circle) into our Table S1

[redacted]

Table 1 of Karamoschos, N. & Tasis, D. Photocatalytic evolution of hydrogen peroxide: a minireview. *Energies* **15**, 6202 (2022).

Table S1 Comparison of H₂O₂ photosynthesis performance.

Table S1 Comparison of H₂O₂ photosynthesis performance.

Photocatalyst	Experimental Conditions		Temperature	Rate ($\mu\text{M h}^{-1}$)	AQY % (420 nm)	Electron Donor	AQY % (Full spectrum)	STH (%)	Ref.
	Gas	Light							
Inorganic Photocatalyst									
CoO_x/Mo-BiVO₄(Ag/Pd)	O ₂	AM 1.5	285K	9700 (irradiation area 4.5 cm ²)	13.1	H ₂ O	3.0	0.73	This work
CoO_x/Mo-BiVO₄/Pd	O ₂	AM 1.5	285K	1425 (irradiation area 1.8 cm ²)	5.8	H ₂ O	1.2	0.29	1
Pd/TiO ₂	Air	AM 1.5	Room temperature	150	-	H ₂ O	-	-	2
rGO/TiO₂/CoP	O ₂	$\lambda > 320$ nm	Room temperature	60	-	H ₂ O	-	-	3
GO	Air	Simulated sunlight	298K	50	-	H ₂ O	-	-	4
rGO/TiO₂/P	O ₂	$\lambda > 320$ nm	Room temperature	30	-	H ₂ O	-	-	3
Au/BiVO ₄	O ₂	$\lambda > 420$ nm	298K	12	0.24	H ₂ O	-	-	5
BiVO ₄	O ₂	$\lambda > 420$ nm	298K	< 0.5	-	H ₂ O	-	-	5
Organic Photocatalyst									
RF/P3HT	O ₂	$\lambda > 300$ nm	333K	3000	11	H ₂ O	-	1.5	6
COF-TpBpy	O ₂	420-700 nm	333K	2000	13.6	H ₂ O	-	1.08	7
RF523	O ₂	420-700 nm	333K	2067	8	H ₂ O	-	0.5	8
Sb-SAPC	O ₂	AM1.5	298K	588	17.6	H ₂ O	-	0.61	9
Co₁/AQ/C₃N₄	O ₂	AM 1.5	Room temperature	62	-	H ₂ O	0.054	0.014	10
g-C₃N₄/BDI50	O ₂	420-500 nm	298K	27.8	2.6	H ₂ O	-	0.13	11
g-C₃N₄/PDI/RGO_{0.05}	O ₂	AM 1.5	298K	24	6.1	H ₂ O	-	0.2	12
g-C₃N₄	O ₂	$\lambda > 420$ nm	Room temperature	175	4.3	H ₂ O	-	0.26	13
ZnPPc/NBCN	O ₂	400-800 nm	Room temperature	11.4	-	H ₂ O	-	-	14
MRFS-7	O ₂	AM 1.5	288K	284.25	-	H ₂ O	-	1.1	15
NH₂-UIO66(Zr)@OPA	O ₂	$\lambda > 420$ nm	Room temperature	9700	-	71 vol% Benzyl Alcohol	-	-	16
cyano-C₃N₄	O ₂	Simulated sunlight	Room temperature	16050	9.58	10 vol% Methanol	-	-	17

We have also revised the manuscript to added more information on the precedents related to the photocatalytic H₂O₂ generation:

Line 37: ‘To tackle the spatiotemporal fluctuations of solar radiation, a promising strategy is to capture solar energy in storable and transportable solar fuels (e.g., H₂ and H₂O₂) by artificial photosynthesis.^{4,5}’

Line 84: ‘Without using any sacrificial agent, the rationally designed BiVO₄ photocatalyst achieved a new record among inorganic photocatalysts (Table S1). *Even comparing with the sacrificial system, CoO_x/BiVO₄(Ag/Pd) has a comparable H₂O₂ production yield (Table S1).* The feasibility of core/shell cocatalyst construction for interfacial-energetics-tuning was further demonstrated on C₃N₄ and TiO₂, demonstrating its high generality for enhancing

charge separation and performance in photosynthesis systems.’

Line 460: ‘[5] Karamoschos, N. & Tasis, D. Photocatalytic evolution of hydrogen peroxide: a minireview. *Energies* **15**, 6202 (2022).’

Comment 3.4 *The authors need to make clear the following: In Figure S5 Which is the third component? There are two components in Figure S7, please assign the second one.*

Response We thank the reviewer for the opportunity to improve the clarity. The unassigned peaks in Figure S5 are the satellite peaks of cobalt. The two components in Figure S7 are Pd⁰ and Pd²⁺. We have updated the figures accordingly:

Figure S5. Co 2p XPS spectra of CoO_x/BiVO₄. The Co 2p_{3/2} peak can be deconvoluted to a Co²⁺ peak at 781.6 eV and a Co³⁺ peak at 780.6 eV. The area of Co 2p_{3/2} peak is twice of that of Co 2p_{1/2} peak and the binding energies of Co³⁺ and Co²⁺ are 780.6 and 781.6 eV, respectively. The other two peaks are the Co satellite peaks.

Figure S7. Pd 3d XPS spectra of CoO_x/BiVO₄/(Ag/Pd). The binding energy Pd 3d_{5/2} and 3d_{3/2} peaks were located at 335.1 eV and 340.6 eV, respectively. The binding energies of Pd⁰ and Pd²⁺ are 335.1 and 337.0 eV, respectively.

Line 112: ‘Prominent Ag 3d and Pd 3d XPS peaks demonstrate the successful loading of metallic Ag/Pd cocatalysts, with the Ag 3d peaks at 368.1 eV and 374.2 eV assigned to the Ag

$3d_{5/2}$ and Ag $3d_{3/2}$, respectively (Fig. S6), and the Pd 3d peaks at 335.1 eV and 340.6 eV assigned to Pd $3d_{5/2}$ and Pd $3d_{3/2}$, respectively (Fig. S7). The minor Pd²⁺ peak at 337.0 eV is attributed to PdO due to the partial oxidation of Pd (Fig. S7). The EDS elemental mapping image shows that CoO_x remained on the {110} facets, while Ag and Pd were selectively loaded on the {010} facets (Fig. 1b).’

Comment 3.5 The authors claim about selective positioning of metal nanoparticles in specific crystallographic planes. I think this statement is rather exaggerating and superficial. EDS images (Fig. 1c,d) show that at the top view we observe islands of metal assemblies, whereas in the side view we get rather dispersed metal nanoparticles. This is more clear in the case of palladium. Thus, the discussion about selective positioning is not standing at all and should be somehow omitted. The authors are suggested to include EDS spectra of Co/Pd and Co/Ag in both views. In addition, please record Raman spectra in the top plane if possible and include in the manuscript. I think that the corresponding cobalt oxide peaks will be visible. Omit the sentence: “SEM image shows that CoOx particles are uniformly distributed across the {110} facets of BiVO4 (Fig. S4a).”

Response Following the reviewer’s suggestion and to provide further evidences for the selective deposition of Pd and Ag, we examined the EDS spectra of CoO_x/BiVO₄/Pd and CoO_x/BiVO₄/Ag. The top view and side view images clearly show that CoO_x was selectively loaded on {110} facet and Ag or Pd was selectively loaded on {010} facet.

Figure S8. Top view (a) and side view (b) of STEM-EDS elemental mapping of Co and Ag particles loaded on BiVO₄.

Figure S9. Top view (a) and side view (b) of STEM-EDS elemental mapping of Co and Pd particles loaded on BiVO₄.

We have also revised the manuscript to clarify the selective deposition of cocatalysts:

Line 119: ‘Notably, the Ag and Pd signals in line profile overlap in distance (Fig. 1b), indicating the co-loading of Ag and Pd nanoparticles on the same sites. The facet-selective loading of Co, Ag and Pd was further confirmed by STEM-EDS mapping from top view and side view of CoO_x/BiVO₄/Ag (Fig. S8) and CoO_x/BiVO₄/Pd (Fig. S9).’

We would like to note that facet-selective loading of cocatalyst through photodeposition is a mature technique and has been realized by multiple groups and studies, for instance:

Liu, T. *et al.* Overall photosynthesis of H₂O₂ by an inorganic semiconductor. *Nat. Commun.* **13**, 1034 (2022).

Li, R. G. *et al.* Spatial separation of photogenerated electrons and holes among {010} and {110} crystal facets of BiVO₄. *Nat. Commun.* **4**, 1432 (2013).

Qi, Y. *et al.* Unraveling of cocatalysts photodeposited selectively on facets of BiVO₄ to boost solar water splitting. *Nat. Commun.* **13**, 484, (2022).

Takata, T. *et al.* Photocatalytic water splitting with a quantum efficiency of almost unity. *Nature* **581**, 411-414 (2020).

Li, R., Han, H., Zhang, F., Wang, D. & Li, C. Highly efficient photocatalysts constructed by rational assembly of dual-cocatalysts separately on different facets of BiVO₄. *Energy Environ. Sci.* **7**, 1369-1376 (2014).

We thank the reviewer for the suggestion on conducting Raman analysis. While it is a cool idea to use Raman spectra to investigate element distribution, the resolution of the

available Raman device (Renishaw inVia Raman Microscope with a 532 nm laser, space resolution 1 μm) is not enough to observe the selective loading of Co particle (< 20 nm) on BiVO_4 (Figure S4). Actually, even the coherent anti-Stokes Raman scattering (CARS) with the highest reported space resolution of 196 nm (Vogt, N. Super-resolution Raman imaging. *Nat. Methods*, **16**, 1202 (2019)), is not eligible in our work. Further, the detection limit of Raman analysis is rather high. For instance, we attempted to investigate the Co, Ag and Pd loading by Raman spectra, and no CoO_x Raman spectra was observed with Co precursor addition up to 1.2 wt%.

Raman spectra of BiVO_4 , $\text{CoO}_x/\text{BiVO}_4/\text{Ag}$, $\text{CoO}_x/\text{BiVO}_4/\text{Pd}$ and $\text{CoO}_x/\text{BiVO}_4/\text{Ag}/\text{Pd}$. The peaks covered by light blue represent the VO_4^{3-} (V-O) in the monoclinic BiVO_4 lattice.

Further, we agree with the reviewer that SEM images alone can not prove the selective loading of CoO_x . The selective loading of Co was mainly confirmed by STEM-EDS mapping and line profile. We have revised the sentence ‘SEM image shows that CoO_x particles are uniformly distributed across the $\{110\}$ facets of BiVO_4 (Fig. S5a).’ accordingly:

Line 99: ‘ CoO_x served to enhance water oxidation and was deposited onto $\{110\}$ facets via photooxidation of Co^{2+} ions ($\text{CoO}_x/\text{BiVO}_4$, Fig. 1a). *SEM image shows that CoO_x particles are loaded on the surface of BiVO_4 (Fig. S4a).* Energy-dispersive X-ray spectroscopy (EDS) elemental mapping and line profile indicate that Co signal on $\{110\}$ facets is much higher than that on $\{110\}$ facets, confirming the selective loading of CoO_x on the $\{110\}$ facets (Fig. 1b).’

Comment 3.6 *There are also similar sentences with analogous meaning which should be omitted. Please perform an additional scavenging experiment (for example, using an hydroxy radical scavenger) in order to assess better the mechanistic path for H_2O_2 evolution. Thus, I do not recommend publication of the submission in the current version.*

Response We have revised the manuscript to improve the clarity and novelty. In addition, we performed additional experiments to assess H_2O_2 evolution path. The H_2O_2 production performance of $\text{CoO}_x/\text{BiVO}_4/\text{Ag}/\text{Pd}$ was evaluated in CH_3OH , NaIO_3 , or under N_2 -purged conditions.

First, when NaIO_3 was added as electron scavenger, H_2O_2 evolution was entirely

inhibited, demonstrating that H_2O_2 was produced by photogenerated electrons. The key role of electrons in H_2O_2 photosynthesis was further demonstrated by improved H_2O_2 production when CH_3OH as hole scavenger was added. In addition, when the system was purged with N_2 to remove O_2 , the H_2O_2 production stopped. The above results together clearly demonstrate that H_2O_2 was produced via O_2 -reduction by photogenerated electrons. This mechanism agrees with the two-electron ORR process widely observed on the surface of Pd (*ACS Catal.* **9**, 626-631 (2019)) (*Appl Catal. B-Environ.* **272**, 119003 (2020)) (*Chem. Eng. J.* **443**, 136429 (2022)) (*Adv Sustain Syst* **6**, 2200144 (2022)). We have added the above results and related discussions in the revised manuscript and SI:

Line 172: ‘For instance, as compared to the low selectivity of BiVO_4/Ag towards H_2O_2 generation (13 %), the H_2O_2 generation selectivity of $\text{BiVO}_4/(\text{Ag}/\text{Pd})$ (81 %) was similar to that of BiVO_4/Pd (85 %, Fig. 2b), suggesting that the Pd shell fully covered Ag core and acted as the dominant H_2O_2 generation sites. H_2O_2 production on $\text{CoO}_x/\text{BiVO}_4/(\text{Pd}/\text{Ag})$ proceeded via a two-electron oxygen-reduction path, as demonstrated by halted H_2O_2 production in the presence of IO_3^- as electron scavenger or under N_2 -purged condition and enhanced H_2O_2 production when methanol as hole scavenger was added (Fig. S14).’

Figure S14. Time courses of photocatalytic H_2O_2 generation in 10%(v/v) CH_3OH , DI water, 10 mM NaIO_3 solution under N_2 condition, and DI water under N_2 condition. Reaction conditions: photocatalyst, 1 mg/mL; 50 ml reaction solution; light source, LED visible light, 300 mW cm^{-2} , $\lambda > 400 \text{ nm}$.

Additional Correction

The “STC” in Figure 2f was corrected to “STH” as below:

Further, Dr. Liu Tian's affiliation was updated since he moved to Suzhou Institute for Advanced Research, University of Science and Technology of China during the preparation of revised manuscript.

Line 6: ‘Suzhou Institute for Advanced Research, University of Science and Technology of China, Suzhou 215000, China’

REVIEWERS' COMMENTS

Reviewer #1 (Remarks to the Author):

The paper has been clarified and improved. I only notice that many paragraphs are too long and suggest a division of the text in a larger number of paragraphs.

Reviewer #3 (Remarks to the Author):

The revised manuscript may be accepted for publication in the current form.

Response to Comments

A general interfacial-energetics-tuning strategy for enhanced artificial photosynthesis

Tian Liu, Zhenhua Pan, Kosaku Kato, Junie Jhon M. Vequizo, Rito Yanagi, Xiaoshan Zheng, Weilai Yu, Akira Yamakata, Baoliang Chen, Shu Hu, Kenji Katayama, Chiheng Chu

Reviewer 1

Comment 1.1 The paper has been clarified and improved. I only notice that many paragraphs are too long and suggest a division of the text in a larger number of paragraphs.

Response We thank the reviewer very much for the thorough review, which help to improve the clarity and quality of our manuscript. We have divided some paragraphs for further clarifying and improving the paper as blow:

Line 115: ‘We further analyzed the structure of the Ag/Pd nanoparticles by scanning transmission electron microscopy-energy dispersive spectrometer (STEM-EDS).’

Line 152: ‘The construction of BiVO₄/Ag junction did not impair the surface reaction selectivity.’

Line 198: ‘As shown in Fig. 3a, CoO_x/BiVO₄/(Ag/Pd) exhibits a longer electron lifetime than CoO_x/BiVO₄/Pd, indicating an enhanced charge-separation process in the former case with a higher ΔV .’

Line 215: ‘CoO_x/BiVO₄/(Ag/Pd) shows a shorter hole lifetime than CoO_x/BiVO₄/Pd, proving that the elevated ΔV enhanced migration of holes from bulk BiVO₄ to CoO_x.’

Line 226: ‘This is a notable result of elevated ΔV that enhanced the charge separation by driving higher extent of holes transferred to CoO_x for HCOOH oxidation.’

Line 277: ‘Based on above results, we propose a general approach for effective interfacial-energetics-tuning (Fig. 5c).’

Line 298: ‘**Cocatalyst deposition.** As-prepared BiVO₄ (0.2 g) was dispersed in 100 ml water, followed by addition of 0.1 mol NaIO₃ and 0.35 mL Co(NO₃)₂ solution (1.5 g/L).’

Reviewer 3

Comment 3.1 *The revised manuscript may be accepted for publication in the current form.*

Response We thank the reviewer very much for the time and effort to improve the quality of our manuscript.

Additional Revision

We have shortened the abstract as blow:

Line 16:

ABSTRACT: The demands for cost-effective solar fuels have triggered extensive research in artificial photosynthesis, yet the efforts in designing high-performance particulate photocatalysts are largely impeded by inefficient charge separation. Because charge separation in a particulate photocatalyst is driven by asymmetric interfacial energetics between its reduction and oxidation sites, enhancing this process demands nanoscale tuning of interfacial energetics on the prerequisite of not impairing the kinetics and selectivity for surface reactions. In this study, we realize this target with a general strategy involving the application of a core/shell type cocatalyst that is demonstrated on various photocatalytic systems. The promising H₂O₂ generation efficiency validate our perspective on tuning interfacial energetics for enhanced charge separation and photosynthesis performance. Particularly, this strategy is highlighted on a BiVO₄ system for overall H₂O₂ photosynthesis with a solar-to-H₂O₂ conversion of 0.73%.